METHODS AND RESOURCES

# Comprehensive glycoproteomics shines new light on the complexity and extent of glycosylation in archaea

Stefan Schulze[1], Friedhelm Pfeiffer[2], Benjamin A. Garcia[3],
Mechthild Pohlschroder[1]*

**1** Department of Biology, University of Pennsylvania, Philadelphia, Pennsylvania, United States of America,
**2** Computational Biology Group, Max Planck Institute of Biochemistry, Martinsried, Germany, **3** Epigenetics
Institute, Department of Biochemistry and Biophysics, Perelman School of Medicine, University of
Pennsylvania, Philadelphia, Pennsylvania, United States of America

* pohlschr@sas.upenn.edu

Pasteur, FRANCE

**Data Availability Statement:** MS raw files of the
dataset generated for this study have been
uploaded to the ProteomeXchange Consortium via
the PRIDE partner repository (https://www.ebi.ac.

## Abstract

Glycosylation is one of the most complex posttranslational protein modifications. Its importance has been established not only for eukaryotes but also for a variety of prokaryotic cellular processes, such as biofilm formation, motility, and mating. However, comprehensive glycoproteomic analyses are largely missing in prokaryotes. Here, we extend the phenotypic characterization of *N*-glycosylation pathway mutants in *Haloferax volcanii* and provide a detailed glycoproteome for this model archaeon through the mass spectrometric analysis of intact glycopeptides. Using in-depth glycoproteomic datasets generated for the wild-type (WT) and mutant strains as well as a reanalysis of datasets within the Archaeal Proteome Project (ArcPP), we identify the largest archaeal glycoproteome described so far. We further show that different *N*-glycosylation pathways can modify the same glycosites under the same culture conditions. The extent and complexity of the *Hfx. volcanii N*-glycoproteome revealed here provide new insights into the roles of *N*-glycosylation in archaeal cell biology.

## Introduction

Posttranslational modifications (PTMs) increase the complexity of the proteome and may affect functional activity, localization, and interactions of proteins. Among the plethora of PTMs, glycosylation is one of the most complex and is encountered in all 3 domains of life, eukaryotes, bacteria, and archaea. The wide distribution and importance of glycosylation in eukaryotes, including its crucial roles in human diseases, have been well established over the past decades [1–4]. However, much less is known about prokaryotic protein glycosylation, as protein glycosylation was long thought to be specific to eukaryotes. Prokaryotic protein glycosylation was first detected in the S-layer glycoprotein (SLG) of the archaeon *Halobacterium salinarum* [5] and soon thereafter also in *Haloferax volcanii* [6]. From there on, more and more prokaryotes, including bacteria, were found to carry glycan modifications of proteins. By now, it is well established that protein glycosylation is common in prokaryotes, and its importance in a variety of biomedically and biotechnologically relevant processes has been revealed such

uk/pride) with the data set identifier PXD021874. Protein database search result files have been added to the ArcPP result repository (https://doi. org/10.5281/zenodo.3724742) while analysis scripts have been included in the ArcPP GitHub repository (https://github.com/arcpp/ArcPP). Source data underlying the graphs in main and supplemental figures are included as S1 Data.

**Funding:** SS was funded by the German Research Foundation (DFG Postdoctoral Fellowship, 398625447). Furthermore, MP, BAG and SS were supported by the National Science Foundation Grant 1817518. BAG also received funding from the National Institute of Health (NIH grant AI118891). The funders had no role in study design, data collection and analysis, decision to publish, or preparation of the manuscript.

**Competing interests:** The authors have declared that no competing interests exist.

**Abbreviations:** ACN, acetonitrile; ArcPP, Archaeal Proteome Project; Cyt, cytosol; DIC, differential inference microscopy; DTT, dithiothreitol; FDR, false discovery rate; GalA, galacturonic acid; Glc, glucose; GlcA, glucuronic acid; Hex, hexose; Man, mannose; MeGlcA, methyl-glucuronic acid; Mem, membrane; MeSQ, methyl-sulfoquinovose; OST, oligosaccharyltransferase; PEP, posterior error probability; PSM, peptide spectrum match; PTM, posttranslational modification; Rha, rhamnose; SHex, sulfated hexose; SLG, S-layer glycoprotein; SN, supernatant; TEAB, triethylammonium bicarbonate; TFA, trifluoroacetic acid; Ths3, thermosome subunit 3; WT, wild-type.

as biofilm formation, pathogenicity, viral protection, and mating [7–10]. However, besides a few abundant surface proteins and appendages, little attention has been paid to which proteins are glycosylated and under which conditions [11–13].

In archaea, glycosylation of SLG, type IV pili, and archaella has been confirmed for various species, including non-haloarchaea like *Methanococcus maripaludis* and *Sulfolobus acidocaldarius* [14–17]. Efforts to identify additional archaeal glycoproteins are limited to the characterization of purified proteins [18], the lectin affinity–based enrichment of glycoproteins [19], and periodic acid–Schiff staining of proteins (as summarized in the ProGlycProt database [11]). Comprehensive glycoproteomic analyses that permit the identification of intact glycopeptides on a proteome-wide scale are missing so far.

Two main types of glycosylation have been encountered in archaea so far: *O*- and *N*-glycosylation. For *O*-glycosylation, which is characterized by the linkage of a glycan to the hydroxyl oxygen of Ser or Thr, only few proteins with short *O*-glycans (mono- and disaccharides) have been identified [5,6], and the archaeal *O*-glycosylation pathway is still elusive. In contrast, *N*-glycosylation pathways, linking glycans to the amide nitrogen of Asn, have been studied in considerable detail in several species, and a plethora of *N*-glycan structures of varying length has been identified [15,17]. Glycan structures are highly variable, even between closely related species [20], and this also holds true for the biosynthetic enzymes that are responsible for the step-by-step assembly of the glycan on its lipid carrier. The assembled lipid-linked oligosaccharide is then transferred onto the protein by 2 general steps: the transfer of the glycan to the extracellular side by a flippase and the attachment to the N-X-S/T sequon of target proteins by an oligosaccharyltransferase (OST).

In *Hfx. volcanii*, 2 distinct *N*-glycosylation pathways have been described so far. One leads to the attachment of a Glc-GlcA-GalA-MeGlcA tetrasaccharide (with Glc, glucose; GlcA, glucuronic acid; GalA, galacturonic acid; MeGlcA, methyl-glucuronic acid), which can be further elongated by a mannose (Man) residue after transfer onto the protein [15,21]. This pathway can be disturbed by deleting *aglB*, the gene coding for the OST. The second pathway results in the transfer of a SHex-Hex-Hex-Rha tetrasaccharide (with SHex, sulfated hexose; Hex, hexose; Rha, rhamnose) and can be disrupted by the deletion of *agl15*, which encodes a flippase [12]. The OST corresponding to this pathway remains to be identified, but transfer of the glycan is independent of AglB. Notably, Agl15-dependent glycans have only been found to modify SLG and only under low-salt growth conditions [12,22]. The enzymes involved in the biosynthesis of this glycan exist, however, under normal growth conditions as revealed by their proteomic identification [23]. Finally, a third *N*-glycan of a more complex type, containing *N*-acetylglucosamine and repeating units of MeSQ-Hex (with MeSQ, methyl-sulfoquinovose), has been found attached to the SLG, but the corresponding biosynthesis pathway awaits identification [24].

A variety of phenotypes have been described for mutants of the AglB-dependent glycosylation pathway. *Hfx. volcanii* Δ*aglB* mutants exhibit a slight growth defect and were observed to be more prone to shedding of the SLG [25,26]. They also appear to be unable to synthesize stable archaella, a phenotype not dependent on SLG instability as specific mutations of glycosites within the archaellins render the cells nonmotile and devoid of detectable archaella [27]. Lack of AglB-dependent glycosylation was also shown to result in aggregation of type IV pili and early induction of microcolony formation [28]. Moreover, a recent study revealed a decreased mating efficiency for Δ*aglB* that was further decreased in a Δ*aglB*/Δ*agl15* double deletion strain [29]. While mating of *Hfx. volcanii* has been shown to be independent of type IV pili and archaella [30], and *N*-glycosylation of the SLG has been suggested to be involved [29], little is known about the many other predicted glycoproteins on the surface of *Hfx. volcanii* that might play key roles in mating and other critical surface-associated functions.

Here, we extended the phenotypic analysis of the Δ*aglB* strain and the less well-characterized Δ*agl15* mutant and carried out a systematic glycoproteomics study to identify as complete of a *Hfx. volcanii* glycoproteome as possible. These studies identified the largest number of glycoproteins yet identified in any archaeon and also revealed that proteins can harbor the so-called low-salt, Agl15-dependent *N*-glycan under normal salt conditions. Furthermore, AglB- and Agl15-dependent *N*-glycosylation occurred not only under the same conditions, but they were also found to be able to modify the same *N*-glycosylation sites.

## Results

### Cell biological assays reveal phenotypes for Δ*aglB* and Δ*agl15* strains under normal salt conditions

In order to further characterize the biological implications of an *aglB* deletion, but especially to gain more insight into the phenotypic effects of an *agl15* deletion, we analyzed the colony morphology and color, the motility, growth, and shape of both Δ*aglB* and Δ*agl15* strains (Fig 1). Colonies that lack Agl15 were darker and smaller than wild-type (WT) colonies (H53), while Δ*aglB* colonies, which were smaller as well, showed the same color as the WT (Fig 1A). Consistent with previous studies, Δ*aglB* mutants were nonmotile [27]. In contrast, Δ*agl15* mutants exhibited the same motility as the WT (Fig 1B).

A growth curve showed that the Δ*aglB* strain exhibited a slightly slower growth than the WT and Δ*agl15* strains during mid-logarithmic phase but reached the same optical density at 600 nm ($OD_{600}$) at stationary phase as the WT (Fig 2A). In contrast, growth of the Δ*agl15* strain was indistinguishable from the WT in early- and mid-logarithmic phase, and a tendency toward a slightly higher $OD_{600}$ than the WT strain in the late-logarithmic growth phase was observed. *Hfx. volcanii* has been described to change from rod-shaped cells in early-logarithmic growth phase to disk-shaped cells in the late-logarithmic growth phase [31–33]. While we observed the same trend for all strains analyzed here, interestingly, the cell shape of both *N*-glycosylation pathway mutants differed from that of the WT strain at different stages of the growth curve (Fig 2B). The Δ*aglB* mutant exhibited a higher ratio of disk- to rod-shaped cells compared to the WT in the early-logarithmic growth phase, and during mid-logarithmic growth, exclusively disk-shaped cells could be found in late-logarithmic Δ*aglB* cultures. In contrast, for Δ*agl15* cells, a tendency toward a higher percentage of rod-shaped cells compared to the WT in the mid- and late-logarithmic growth phase was observed. A quantitative analysis

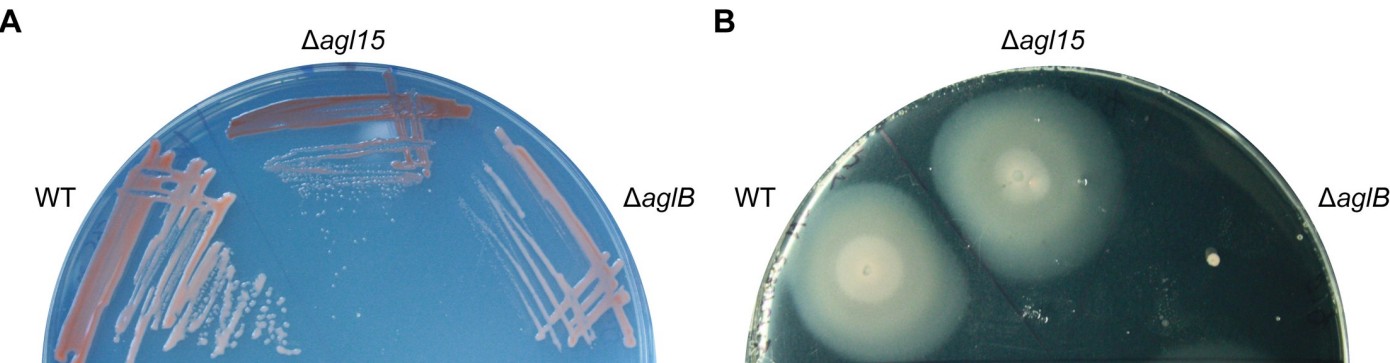

**Fig 1. Colony morphology and motility phenotypes of *N*-glycosylation pathway mutants. (A)** WT, Δ*agl15*, and Δ*aglB* strains were streaked out on Hv-Cab plates with 1.5% agar. Colonies of Δ*agl15* and Δ*aglB* mutants appear smaller, and for the Δ*agl15* mutant, also darker than WT colonies. **(B)** WT, Δ*agl15*, and Δ*aglB* strains were stabbed into motility plates (Hv-Cab with 0.35% agar) and imaged after 5 days of incubation. While the WT and Δ*agl15* strains show normal motility, the Δ*aglB* mutant is nonmotile. WT, wild-type.

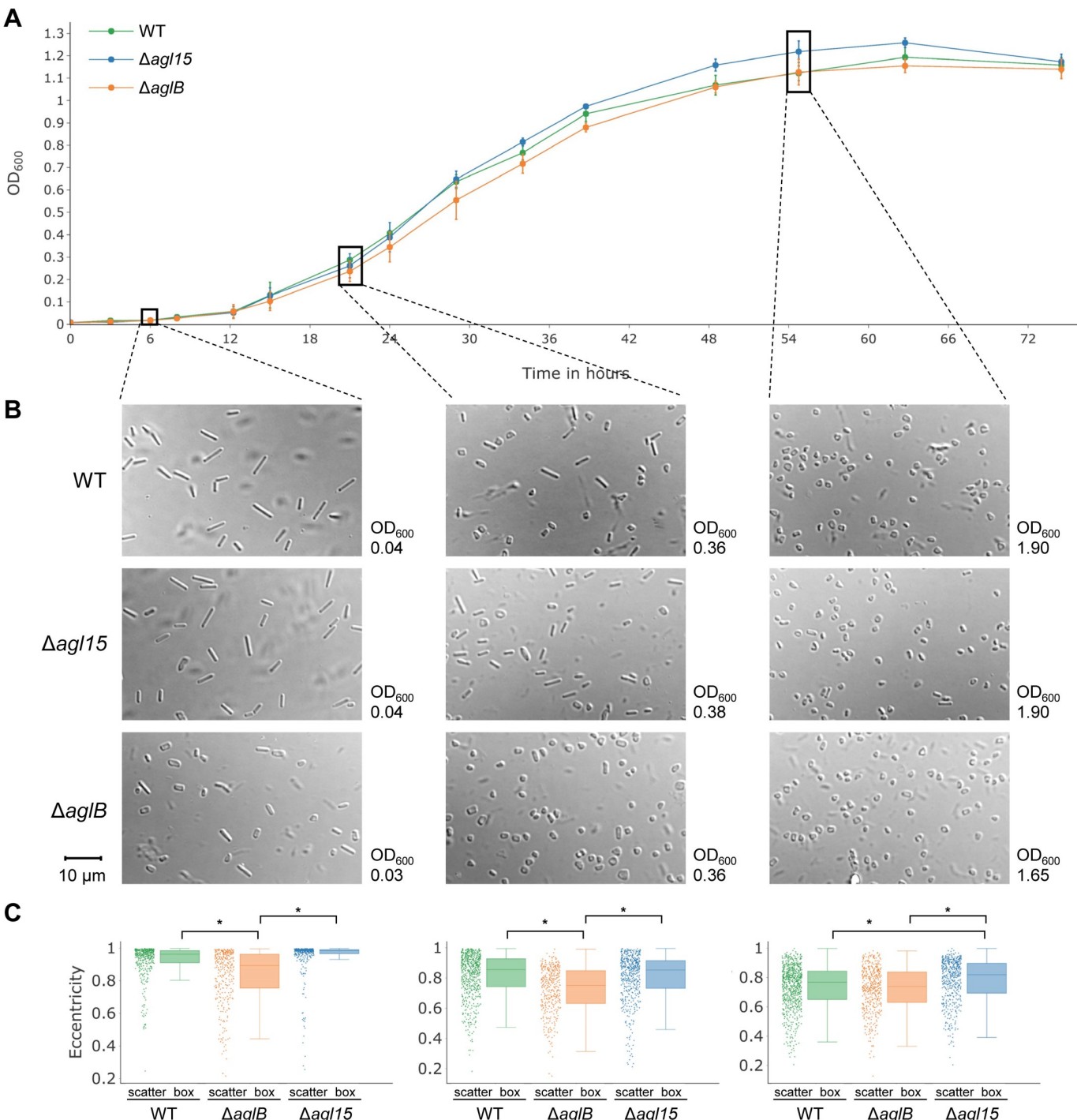

**Fig 2. Growth and cell shape phenotypes of *N*-glycosylation pathway mutants. (A)** The growth of WT (green), Δ*agl15* (blue), and Δ*aglB* (orange) strains was analyzed by measuring the $OD_{600}$ of cultures over the course of 3 days. Data points and error bars represent the mean and standard deviation of 3 biological replicates. **(B)** Samples of early- (left), mid- (middle), and late-logarithmic (right) growth phase cultures for each strain were imaged using DIC microscopy. The majority of WT cells in the early-logarithmic growth phase is rod-shaped, and the ratio of rod- to disk-shaped cells decreases over time. A higher ratio of rod- to disk-shaped cells was observed in mid- and late-logarithmic growth phase for the Δ*agl15* mutant in comparison to the WT, while almost no rod-shaped cells were visible for the Δ*aglB* strain. It should be noted that $OD_{600}$ measurements for cell shape samples were performed with a path length of 1.5 cm, while growth curve measurements were performed in 96-well plates with 250 µl of culture. Images in A, B, and C (bottom) are representative for at least 4 biological replicates. The scale bar (bottom left) indicates 10 µm and applies to all DIC microscopy images. **(C)** A quantitative cell shape analysis was performed using CellProfiler [34]. Boxplots for each growth phase depict the eccentricity of cells for each strain across 3 biological replicates. Individual cells are shown as scatter plot, while the center line, box limits, and whiskers of each boxplot represent the median, upper/lower quartiles, and 1.5× interquartile range, respectively. A *t* test with

subsequent Benjamini–Hochberg correction for multiple testing was performed, and statistically significant differences ($p < 10^{-5}$) are indicated by asterisks ($^*$). The underlying source data for A and C can be found in S1 Data. DIC, differential inference microscopy; WT, wild-type.

using CellProfiler confirmed that the cell shape of Δ*aglB* and Δ*agl15* cultures differed significantly ($p < 10^{-5}$) from WT cultures at early- and late-logarithmic growth phases, respectively (Fig 2C).

In summary, these results revealed multiple biological phenotypes for *Hfx. volcanii* strains lacking AglB- or Agl15-dependent *N*-glycosylation, indicating that both pathways are involved in a variety of cellular processes. Since all assays were performed using media with optimal salt concentrations (18% salt water corresponding to 2.7 M salts), this suggests that, consistent with the Archaeal Proteome Project (ArcPP) data, Agl15-dependent glycosylation is not restricted to low-salt conditions. Importantly, the phenotypes of both deletion mutants were distinct from each other, indicating differing roles in the same cellular processes.

The diversity and multiplicity of the biological effects caused by the interference with *N*-glycosylation pathways may involve additional *N*-glycoproteins besides the well-characterized SLG, archaellins, and pilins. Thus, in the following we aimed to provide a more complete picture about the *Hfx. volcanii* glycoproteome.

## Glycoprotein staining reveals minimal differences between the WT, Δ*aglB*, and Δ*agl15* strains

In order to determine which *Hfx. volcanii* proteins can be glycosylated, we devised a comprehensive glycoproteomic analysis of samples taken under salt conditions optimal for the growth of this haloarchaeon. This analysis was performed for the WT, Δ*agl15*, and Δ*aglB* strains, thus including negative controls for each described *N*-glycosylation pathway, allowing to control for false positive identifications. Cells were fractionated into cytosol (Cyt), membrane (Mem), and culture supernatant (SN) to achieve a high proteome coverage by reducing the sample complexity and to obtain information about the cellular localization of the identified glycoproteins. Separation of these fractions by LDS-PAGE followed by Coomassie staining demonstrated that for the Δ*aglB* mutant, a larger portion of the SLG was shed into the SN than for the WT and Δ*agl15* strains (S1A Fig). This is in line with previous reports indicating a reduced S-layer stability on cells lacking AglB [25,26]. Besides this, a similar banding pattern was observed for all strains. Similarly, a glycoprotein stain showed no apparent differences between the strains either (S1B Fig). It may be surprising that the disruption of an *N*-glycosylation pathway does not affect the banding pattern of glycoproteins. This observation is, however, in line with previous reports using the conceptually similar periodic acid–Schiff staining, which showed only a small decrease in signal intensity for SLG from the Δ*aglB* mutant [29]. Similar signal intensities for the Pro-Q Emerald 300 glycoprotein staining employed here may indicate the presence of Agl15-dependent glycosylation in the Δ*aglB* mutant and vice versa. Alternatively, *O*-glycosylation or additional *N*-glycosylation through unknown biosynthesis pathways [24] could mask differences in AglB/Agl15-dependent *N*-glycosylation patterns as well. Finally, while unspecific staining of non-glycosylated proteins can be observed, it has been accounted for by using non-glycosylated control samples (S1B Fig).

## Proteomic analysis of *Hfx. volcanii* reveals the largest archaeal glycoproteome described to date

For the mass spectrometric analysis of the fractions obtained from WT, Δ*agl15*, and Δ*aglB* strains, samples from cultures in mid- and late-logarithmic growth phases were mixed using

equal culture volumes. While quantitative analyses are beyond the scope of this work, combining these samples increased the coverage of glycoproteins that are potentially expressed in different growth conditions. This allowed for a comprehensive assessment of glycosylation in *Hfx. volcanii* and a qualitative comparison between the analyzed strains. In order to identify intact glycopeptides, we used the combination of multiple protein database search engines as previously described within the ArcPP [23]. Importantly, the glycans of each step in the 2 known *N*-glycosylation pathways of *Hfx. volcanii* (in total 10 glycans) [12,15], as well as di-hexose as the only described *O*-glycosylation for *Hfx. volcanii* so far [6], were included in the search as potential modifications. Since this substantial expansion of the search space could lead to an increase in false positive identifications, a stringent filtering of search results was performed. Besides a posterior error probability (PEP) ≤1% on the peptide spectrum match (PSM) level, (glyco-)peptides were required to be identified by at least 2 spectra and to have a peptide false discovery rate (FDR) ≤1%.

Protein database search engines only take into account the fragmentation of the peptide backbone. Therefore, spectra corresponding to glycopeptide identifications were searched subsequently for glycopeptide-specific ions. In analogy to the b- and y-ion series for the peptide backbone, which are due to preferential cleavage of the peptide bond, glycans are frequently cleaved at the interglycosidic bond [35]. This leads to a Y-ion series (representing the peptide plus the remainder of the glycan, the agylcan version being Y0) and a B-ion series (or oxonium-ion series) representing the cleaved-off part of the glycan (the liberated terminal sugar being B1). For safe glycopeptide identifications, the detection of at least 1 B-ion and 2 Y-ions was considered mandatory. While stringent criteria are expected to strongly reduce the likelihood of false positive identifications, it should be noted that they become more easily satisfied when glycans are longer. Since the glycans of *O*-glycopeptides are very short, we focused in this work on the detailed analysis of *N*-glycopeptides. However, *O*-glycopeptides were nevertheless identified (see below).

Through this extensive glycoproteomic approach, a total of 194 *N*-glycopeptides, spanning 102 *N*-glycosites, were identified for 39 *N*-glycoproteins (Fig 3, Table 1). These results demonstrate that *N*-glycosylation occurs in *Hfx. volcanii* to a much higher extent than previously known, as only 5 *N*-glycoproteins were described before (SLG, ArlA1, ArlA2, PilA1, and PilA2 [6,27,28]). Notably, the reliability of this search and filtering approach is highlighted by the lack of AglB- and Agl15-dependent glycans in the respective deletion mutants. Furthermore, all *N*-glycoproteins identified in this dataset (PXD021874) are predicted to be secreted or transmembrane proteins (Table 1), which was expected since the OST is located on the extracellular side of the plasma membrane. Exceptions from other datasets (e.g., Ths3 with predicted cytosolic localization, identified in dataset PXD011056) are described and discussed below. Overall, our approach allowed for the reliable large-scale identification of archaeal *N*-glycopeptides.

The vast majority of identified *N*-glycopeptides was found in both the WT and Δ*agl15* strains (Fig 3A and 3B), suggesting a high similarity of the *N*-glycoproteome between the 2 strains. Furthermore, almost all identified *N*-glycopeptides harbor AglB-dependent glycans (Table 1). This is in line with previous reports establishing the AglB-dependent pathway as the predominant *N*-glycosylation pathway in *Hfx. volcanii* [12,36]. However, interestingly, Agl15-dependent *N*-glycopeptides were identified in samples from the WT strain, revealing that Agl15-dependent glycosylation does not only occur under low-salt conditions. The peptides harboring Agl15-dependent glycans correspond to the SLG, which has previously been studied extensively. Asn residues at positions 13, 83, 274, and 279 had been shown to be modified with AglB-dependent glycans [21], while N498 was found harboring an Agl15-dependent glycan [36] and N732 carrying a glycan from an as yet unknown *N*-glycosylation pathway [24]

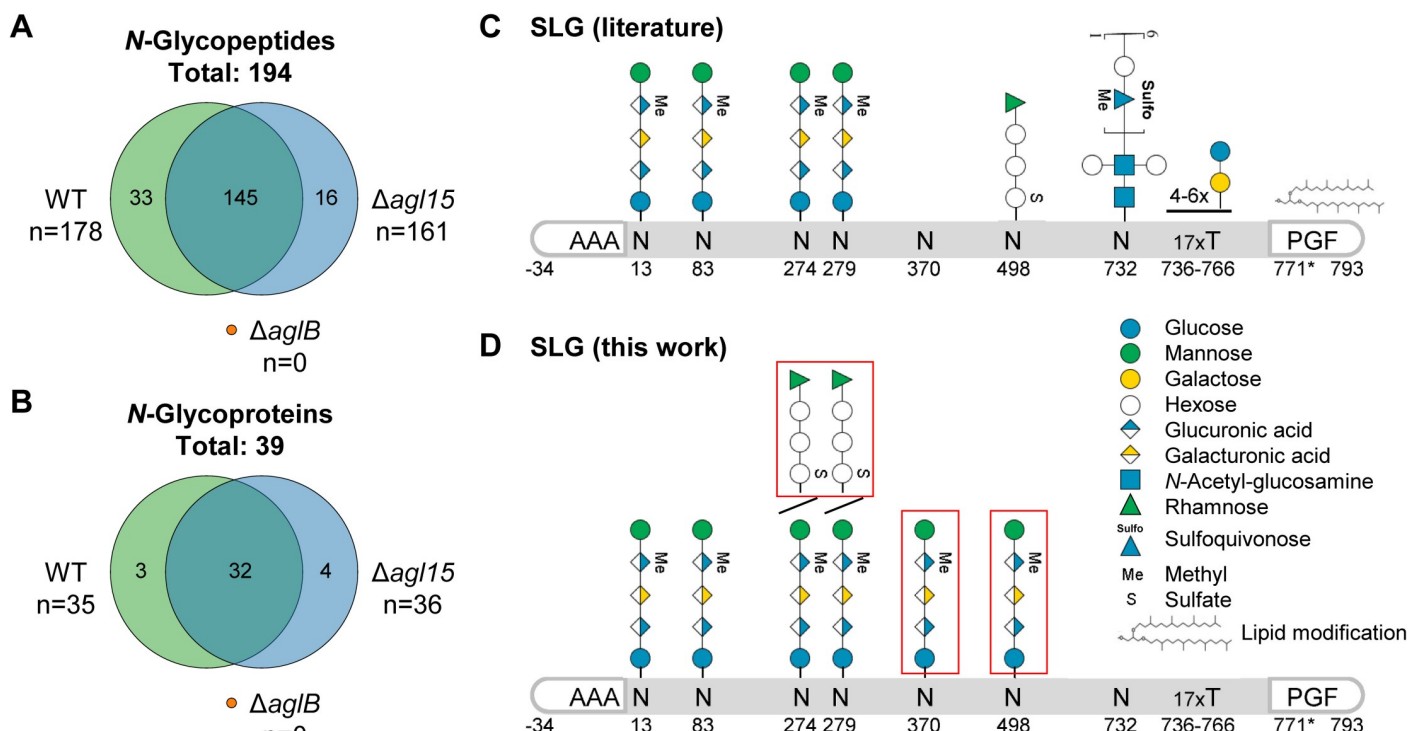

**Fig 3. Glycoproteomic analysis of *Hfx. volcanii* reveals concurrent AglB- and Agl15-dependent *N*-glycosylation.** Cellular fractions of WT, Δ*agl15*, and Δ*aglB* strains were analyzed by LC–MS/MS. The overlap of identified *N*-glycopeptides (**A**) and *N*-glycoproteins (**B**) between the 3 strains is represented as Venn diagrams. No *N*-glycopeptides were identified for Δ*aglB*. Since *Hfx. volcanii* SLG has been extensively studied previously, the glycosites (numbers indicating the amino acid position after signal peptide cleavage) and corresponding glycans that have been described so far ([6,21,24,36], figure adapted from [37]) are depicted schematically (**C**). *N*-glycosites and corresponding *N*-glycans that have been identified in this study are shown schematically (**D**). For some *N*-glycosites, multiple *N*-glycans were identified, indicated by diagonal lines. Furthermore, *N*-glycans that have not been identified previously are highlighted by red boxes. It should be noted that *N*-glycopeptides with shorter versions of the AglB-dependent pentasaccharide have been described previously and identified here as well, but are not depicted separately. LC–MS/MS, liquid chromatography tandem mass spectrometry.

(Fig 3C). Our analysis now identified an additional *N*-glycosite, N370, harboring AglB-dependent glycans. Furthermore, N274 and N279 were found to be modified with AglB- as well as Agl15-dependent glycans, while N498 carried AglB-dependent glycosylation (Fig 3D, S2 Fig). Together, these results show not only that Agl15-dependent *N*-glycosylation can occur under the same conditions as AglB-dependent *N*-glycosylation but also that both pathways can modify the same *N*-glycosites.

Peptides containing both *N*-glycosites N274 and N279 were identified harboring either 2 AglB- or 2 Agl15-dependent *N*-glycans. Unfortunately, we could not assess whether peptides containing 2 *N*-glycosites could be modified with 1 AglB- and 1 Agl15-dependent *N*-glycan, since the employed protein database search engines do not support the simultaneous search for 2 different modifications of the same amino acid. However, mass shifts corresponding to a combination of AglB- and Agl15-dependent *N*-glycosylation could not be found in results from an open modification search [38].

## Glycoproteomic analysis of the ArcPP datasets further extends glycoprotein identifications

After establishing that our workflow resulted in the reliable identification of glycopeptides, we extended our glycoproteomics analysis to suitable datasets from the ArcPP [23], while at the same time including our newly generated proteomic dataset in ArcPP. Following the same

**Table 1. Summary of identified *N*-glycoproteins.**

| HVO ID | Name | Description | *N*-glycosites | *N*-glycan type(s) | PSMs | Dataset(s) | Predicted processing |
|---|---|---|---|---|---|---|---|
| HVO_0307 | - | Conserved hypothetical protein | 3 | AglB | 201 | PXD021874; PXD006877; PXD010824; PXD011012; PXD011050; PXD011056 | Sec (SPI) |
| HVO_0504 | - | DUF192 family protein | 1 | AglB | 32 | PXD021874 | Sec (SPI) |
| HVO_0778 | Ths3 | Thermosome subunit 3 | 1 | Agl15 | 4 | PXD011056 | Cyt |
| HVO_0892 | NosD | ABC-type transport system periplasmic substrate-binding protein (probable substrate copper) | 4 | AglB | 193 | PXD021874; PXD011012; PXD011218 | Sec (SPI) |
| HVO_0972 | PilA1 | Pilin PilA | 3 | AglB | 424 | PXD021874; PXD006877; PXD009116; PXD010824; PXD011012; PXD011015; PXD011050; PXD011056; PXD014974 | Pil (SPIII) |
| HVO_1014 | CoxB1 | Cox-type terminal oxidase subunit II | 2 | AglB | 13 | PXD021874 | Sec (SPI) |
| HVO_1030 | - | DUF4382 domain protein | 2 | AglB | 55 | PXD021874; PXD011050 | Sec (lipobox) |
| HVO_1176 | - | Conserved hypothetical protein | 2 | AglB | 54 | PXD021874; PXD009116; PXD011218; PXD013046; PXD014974 | Sec (lipobox) |
| HVO_1210 | ArlA1 | Archaellin A1 | 2 | AglB | 63 | PXD021874; PXD011012; PXD011015; PXD011050 | Pil (SPIII) |
| HVO_1211 | ArlA2 | Archaellin A2 | 1 | AglB | 6 | PXD021874; PXD011012; PXD011050 | Pil (SPIII) |
| HVO_1259 | - | Conserved hypothetical protein | 2 | AglB | 47 | PXD021874 | TM N-term |
| HVO_1530 | AglB | Dolichyl-monophosphooligosaccharide—protein glycotransferase AglB | 2 | AglB | 256 | PXD021874; PXD010824; PXD011012; PXD011050; PXD011056; PXD014974 | $\geq$ 2 TM |
| HVO_1624 | - | Conserved hypothetical protein | 1 | AglB | 3 | PXD021874 | Tat (lipobox) |
| HVO_1673 | - | Conserved hypothetical protein | 2 | AglB | 116 | PXD021874; PXD009116; PXD011012; PXD011050; PXD011218 | Sec (lipobox) |
| HVO_1749 | - | Conserved hypothetical protein | 2 | AglB | 153 | PXD021874; PXD010824; PXD011012; PXD011050; PXD011056 | Pil (SPIII) |
| HVO_1802 | - | Peptidase M10 family protein | 1 | AglB | 2 | PXD021874 | Sec (lipobox) |
| HVO_1806 | - | Conserved hypothetical protein | 1 | AglB | 8 | PXD021874 | Sec (lipobox) |
| HVO_1870 | - | M50 family metalloprotease | 2 | AglB | 97 | PXD021874; PXD010824; PXD011050 | $\geq$ 2 TM |
| HVO_1944 | - | Probable transmembrane glycoprotein/HTH domain protein | 1 | AglB | 16 | PXD021874 | Sec (SPI) |
| HVO_1945 | - | Conserved hypothetical protein | 4 | AglB | 183 | PXD021874 | Tat (SPI) |
| HVO_1976 | SecD | Protein-export membrane protein SecD | 2 | AglB | 33 | PXD021874; PXD010824; PXD011012; PXD013046; PXD014974 | Sec (SPI) |
| HVO_1988 | - | GATase domain protein | 1 | AglB | 31 | PXD021874 | Sec (SPI) |
| HVO_2062 | PilA2 | Pilin PilA | 2 | AglB | 56 | PXD021874; PXD011012; PXD011015; PXD011050 | Pil (SPIII)* |
| HVO_2066 | - | Conserved hypothetical protein | 1 | AglB, Agl15 | 27 | PXD006877 | Sec (SPI) |
| HVO_2070 | - | Conserved hypothetical protein | 2 | AglB | 100 | PXD021874; PXD011012; PXD011015; PXD013046 | Sec (SPI) |
| HVO_2071 | - | Probable secreted glycoprotein | 4 | AglB | 134 | PXD021874; PXD006877; PXD011012; PXD011050 | Sec (SPI) |
| HVO_2072 | SLG | SLG | 6 | AglB, Agl15 | 2364 | PXD021874; PXD006877; PXD007061; PXD009116; PXD010824; PXD011012; PXD011015; PXD011050; PXD011056; PXD011218; PXD013046; PXD014974 | Sec (SPI)*, ArtA |
| HVO_2074 | - | Probable secreted glycoprotein | 1 | AglB | 2 | PXD021874; PXD011050 | Sec (SPI) |
| HVO_2076 | - | Probable secreted glycoprotein (nonfunctional) | 3 | AglB | 42 | PXD021874 | Sec (SPI) |
| HVO_2081 | - | Pectin lyase domain protein | 3 | AglB | 52 | PXD021874; PXD011050 | Sec (SPI) |

(*Continued*)

**Table 1.** (Continued)

| HVO ID | Name | Description | N-glycosites | N-glycan type(s) | PSMs | Dataset(s) | Predicted processing |
|---|---|---|---|---|---|---|---|
| HVO_2082 | - | Conserved hypothetical protein | 2 | AglB | 117 | PXD021874; PXD006877; PXD011012; PXD011050 | Sec (SPI) |
| HVO_2084 | - | ABC-type transport system permease protein (probable substrate macrolides) | 1 | AglB | 50 | PXD021874 | ≥ 2 TM |
| HVO_2160 | - | Probable secreted glycoprotein | 19 | AglB | 1561 | PXD021874; PXD006877; PXD007061; PXD009116; PXD010824; PXD011012; PXD011050; PXD011056; PXD011218; PXD013046; PXD014974 | Sec (SPI), ArtA |
| HVO_2161 | - | Probable secreted glycoprotein | 1 | AglB | 5 | PXD021874; PXD011012 | Sec (SPI) |
| HVO_2167 | - | Conserved hypothetical protein | 1 | AglB | 12 | PXD021874; PXD006877 | Sec (SPI) |
| HVO_2172 | - | Conserved hypothetical protein | 3 | AglB | 82 | PXD021874 | Sec (SPI) |
| HVO_2173 | - | DUF1616 family protein | 4 | AglB | 83 | PXD021874 | ≥ 2 TM |
| HVO_2533 | - | Conserved hypothetical protein | 5 | AglB | 103 | PXD021874; PXD007061; PXD009116; PXD011012; PXD011056; PXD011218; PXD013046 | Sec (SPI), ArtA |
| HVO_2535 | - | Conserved hypothetical protein | 1 | AglB | 2 | PXD021874 | Sec (lipobox) |
| HVO_2634 | - | Conserved hypothetical protein | 1 | AglB | 40 | PXD011012 | Sec (SPI) |
| HVO_A0039 | - | Conserved hypothetical protein | 2 | AglB | 93 | PXD021874; PXD007061; PXD013046 | Sec (SPI) |
| HVO_A0466 | - | Conserved hypothetical protein | 1 | AglB | 9 | PXD021874 | Sec (SPI) |
| HVO_A0499 | - | Conserved hypothetical protein | 2 | AglB | 27 | PXD021874 | Sec (SPI) |
| HVO_B0194 | - | LppX domain protein | 4 | AglB | 46 | PXD021874 | Sec (SPI) |
| HVO_C0054 | - | Hypothetical protein | 3 | AglB | 29 | PXD021874 | Tat (SPI) |

For each protein that was identified to be *N*-glycosylated in this study, the HVO ID, name, and description are given together with the number of identified *N*-glycosites, the *N*-glycan type(s), the number of corresponding PSMs, and the dataset(s) in which it was identified to be *N*-glycosylated, and the predicted processing. Datasets are given as PRIDE IDs, and it should be noted that PXD021874 corresponds to the dataset generated for this manuscript, while all other PRIDE IDs correspond to datasets of the ArcPP.

\*The processing for this entry has been corrected upon manual curation.

ArcPP, Archaeal Proteome Project; ArtA, archaeosortase A substrate; Cyt, cytosolic; HTH, helix–turn–helix; lipobox, conserved cleavage site motif for lipoproteins; Pil, type IV pilin pathway; PSM, peptide spectrum match; Sec, Sec pathway; SLG, S-layer glycoprotein; SPI, signal peptidase I; SPIII, signal peptidase III; Tat, twin arginine translocation pathway; TM, transmembrane domain.

filtering criteria as described above (except for the mandatory detection of oxonium-ions in datasets that did not include low m/z ranges in MS2 spectra), a total of 233 *N*-glycopeptides spanning 114 *N*-glycosites corresponding to 45 *N*-glycoproteins were identified (Fig 4A, Table 1). This represents a remarkable increase in the number of identified *N*-glycoproteins and *N*-peptides of 15% and 20%, respectively, compared to the separate analysis of PXD021874. These results highlight the value of such a combined reanalysis for the identification of complex PTMs like glycosylation. Furthermore, it allows for comparisons between the datasets. When taking into account all peptides, independent of their glycosylation, the vast majority of *N*-glycoproteins could be identified in all datasets that analyzed whole proteomes (Fig 4B). However, most *N*-glycopeptides were only identified in the dataset generated for this study (PXD021874). Datasets with more than the median 8 *N*-glycoprotein identifications (PXD021874, PXD011015, and PXD011050) used both trypsin and GluC for protein digestion before MS measurements (Fig 4A), indicating that the cleavage by GluC aided in the detection of *N*-glycopeptides. Further increases in *N*-glycopeptide identifications between PXD021874 and PXD011015/PXD011050 can likely be attributed to cell fractionation, peptide fragmentation using stepped collisional energies, and high sensitivity of the employed instrument.

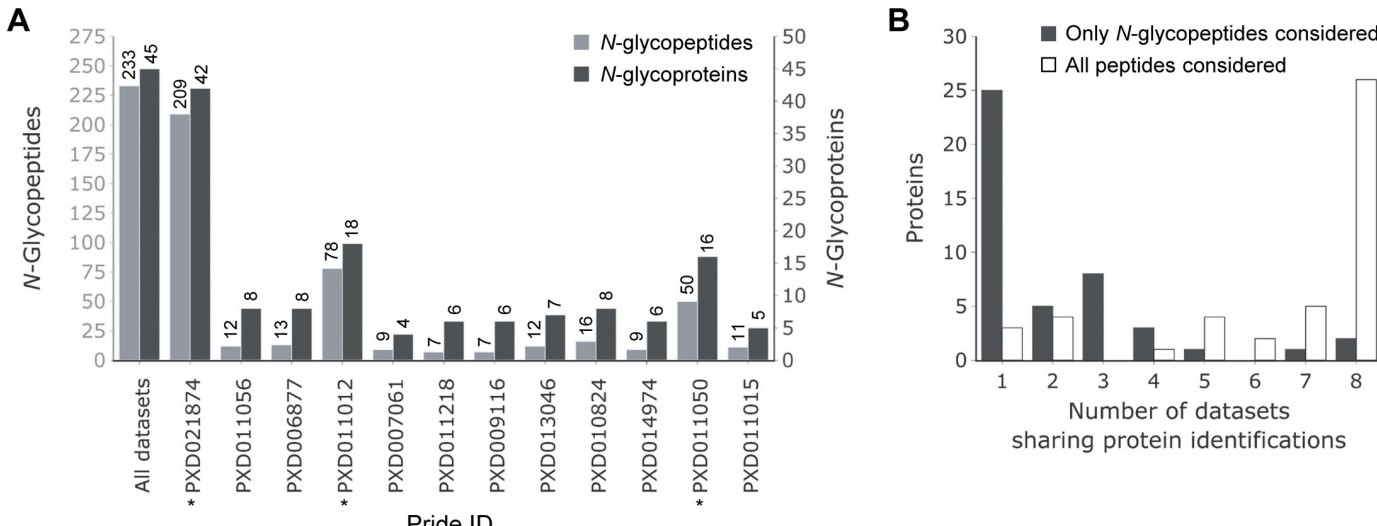

**Fig 4. Glycoproteomic analysis of ArcPP datasets extends the *Hfx. volcanii* N-glycoproteome.** Datasets included in the ArcPP, supplemented by dataset PXD021874 from the current study, were reanalyzed including *Hfx. volcanii* glycans as potential modifications. **(A)** The number of identified *N*-glycopeptides (light gray) and *N*-glycoproteins (dark gray) for each dataset is shown as a barplot (sorted by the total number of identified proteins; see S2 Fig). Datasets, for which the enzymatic digest was performed with trypsin as well as GluC, are marked with an asterisk (*). **(B)** For each identified *N*-glycoprotein, the number of whole proteome datasets that share this identification was determined. The number of *N*-glycoproteins identified in the given number of datasets is represented as a barplot. This analysis was performed taking into account either only *N*-glycopeptides (dark gray) or all peptides identified for a protein that was determined to be *N*-glycosylated in any of the datasets (white). The underlying source data for A and B can be found in S1 Data. ArcPP, Archaeal Proteome Project.

Notably, taking together glycosylated and non-glycosylated peptides and proteins, PXD021874 represents the most comprehensive *Hfx. volcanii* proteome dataset within the ArcPP (S3 Fig). Consequently, the inclusion of PXD021874 in the ArcPP increased the number of identified *Hfx. volcanii* proteins from 2,912 to 3,036 (72% and 74.5% of the theoretical proteome, respectively), with a median protein sequence coverage of 54%.

The identification of additional *N*-glycopeptides confirmed and extended our findings about the concurrence of AglB- and Agl15-dependent glycosylation. The SLG peptide containing *N*-glycosites N274 and N279 was found modified with AglB- and Agl15-dependent glycans in an additional dataset (PXD011012, Table 1). Furthermore, both *N*-glycan types were found on *N*-glycopeptides corresponding to a second protein, HVO_2066 (PXD006877; Table 1). However, the vast majority of identified *N*-glycopeptides harbored exclusively AglB-dependent *N*-glycans, and only 1 *N*-glycopeptide with exclusively Agl15-dependent *N*-glycans was found. Surprisingly, the latter originates from the thermosome subunit 3 (Ths3), which is the only identified *N*-glycoprotein with a predicted cytosolic localization (Table 1). The identification of Ths3 as an *N*-glycoprotein, which was found exclusively in dataset PXD011056, is well supported by 4 corresponding MS2 spectra through almost complete b- and y-ion series as well as a complete series of glycan-specific Y-ions (S4 Fig). Therefore, this is unlikely to represent a false positive identification. While it cannot be excluded that Agl15-dependent *N*-glycosylation occurs in the Cyt as well and that the close interaction of thermosomes with other proteins, potentially including OSTs, could result in coincidental *N*-glycosylation, it is worth noting that approximately the same number of PSMs and peptides of Ths3 were identified in membrane fractions as in Cyt fractions (see Data Availability Statement for deposited search results). Furthermore, thermosome subunits were described to be surface-associated in *Sulfolobus shibatae* [39], and the identification of thermosome subunits in lectin affinity enrichments from *Methanosarcina mazei* and *Methanosarcina acetivorans* [40] further supports the surface association and *N*-glycosylation of Ths3. Thus, our identification of *N*-glycosylated peptides of Ths3

provides additional evidence for the hypothesis that thermosome subunits are not restricted to a cytosolic localization.

## Unknown functions but high degree of phylogenetic conservation within Halobacteria for most identified *N*-glycoproteins

After revealing this remarkably extensive and complex *N*-glycoproteome of *Hfx. volcanii*, we continued our study with a phylogenetic analysis of all 45 identified *N*-glycoproteins. While the functions and *N*-glycosylation of SLG, ArlA1, ArlA2, PilA1, and PilA2 have been established [26–28], we were able to identify additional *N*-glycoproteins with known functions that previously were not considered to be *N*-glycosylated. Interestingly, besides SecD (membrane protein involved in protein export), this includes the OST AglB, which itself is *N*-glycosylated in an AglB-dependent manner. Moreover, for a few other *N*-glycoproteins, functions can be inferred from characterized homologs and/or conserved domains: CoxB1, a subunit of a cox-type terminal oxidase (HVO_1014); a thermosome subunit (Ths3, HVO_0778); 1 substrate-binding protein and 1 permease component of ABC-type transport systems (HVO_0892 and HVO_2084); as well as an M10 family metallopeptidase and an M50 family metalloprotease (HVO_1802 and HVO_1870).

However, a large part of the identified *N*-glycoproteins is uncharacterized (two-thirds of *N*-glycoproteins compared to one-third of the theoretical proteome), lacking characterized homologs, that are closely enough related to imply similar functions. The vast majority of these proteins is even devoid of InterPro domain assignments, and, therefore, belongs to the genomic dark matter. Therefore, for all identified *N*-glycoproteins, the current annotation has been evaluated in order to inspect if current knowledge allows the assignment of protein functions (S1 Table). During this analysis, proteins were also subjected to gene synteny analysis using SyntTax [41]. Furthermore, all proteins were categorized with respect to the taxonomic range of their orthologs employing OrthoDB (S1 Table) [42]. Within the taxonomic range, the number of genera coding for an ortholog was determined, revealing that for most *N*-glycoproteins, only a few orthologs were found outside the class Halobacteria (Fig 5). However, results indicated a higher degree of conservation within the Halobacteria. Therefore, within this class, the same phylogenetic analysis was done at the species level, and orthologs for the majority of species and *N*-glycoproteins were identified (S5 Fig).

Notably, several *N*-glycosylated proteins are encoded in genomic vicinity to each other (S1 Table). Besides the known cluster of *arlA1/arlA2*, this includes a region spanning from *hvo_2062* (*pilA2*) to *hvo_2084* (ABC-type transport system component), which encode for 10 *N*-glycoproteins, one of which is SLG (*hvo_2072*). Furthermore, a cluster from *hvo_2160* to *hvo_2173* encodes 5 *N*-glycoproteins, with HVO_2160 being the largest *Hfx. volcanii* protein that also contains the highest number of confirmed *N*-glycosites (19). However, the reason for this clustering remains to be elucidated, as the function of most proteins encoded within these clusters is unknown. This is also the case for 3 additional gene clusters corresponding to *N*-glycoproteins of unknown function.

## Evidence for *N*-glycosylation at noncanonical *N*-glycosites

In vitro experiments previously suggested that AglB-dependent *N*-glycosylation in archaea may not be strictly specific to the canonical N-X-S/T sequon [43]. We therefore checked for noncanonical *N*-glycosites by omitting the requirement for N-X-S/T sequons in the final filtering of glycopeptide identifications. Indeed, this revealed the confident identification of 5 *N*-glycopetides corresponding to 3 noncanonical *N*-glycosites from 3 different proteins (Table 2). Interestingly, this includes the heavily *N*-glycosylated HVO_2160 as well as PilA6 (S6 Fig),

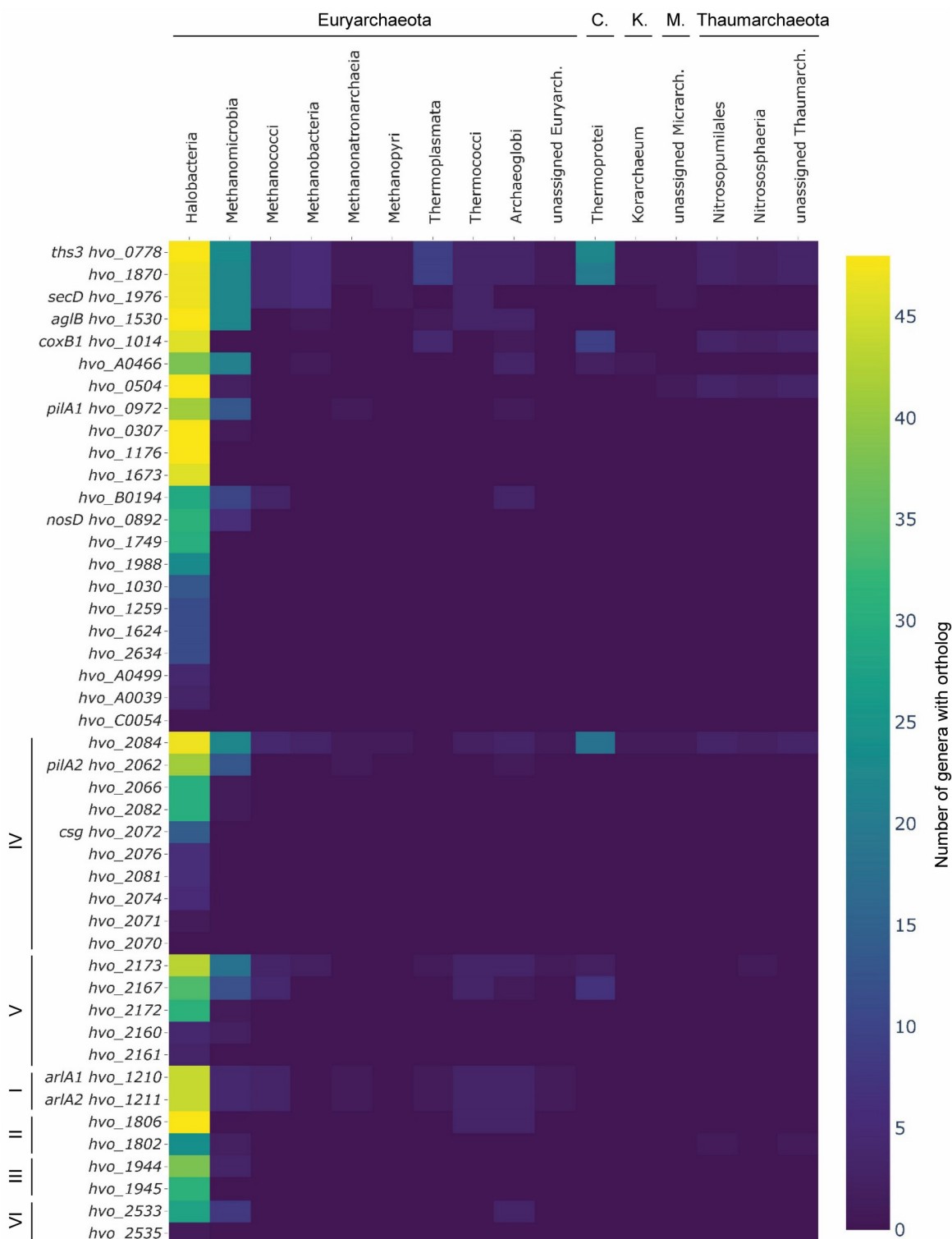

**Fig 5. Phylogenetic analysis of identified *N*-glycoproteins reveals conservation within Halobacteria.** The HVO IDs of all identified *N*-glycoproteins have been subjected to OrthoDB analysis in order to determine orthologues across the archaeal domain. For each *N*-glycoprotein encoding gene, the number of genera within the different archaeal taxonomic classes is given as a heatmap ranging in color from yellow (48 genera with orthologous proteins) to purple (0 genus). Genes were sorted by the overall number of genera with orthologous proteins, while clusters with multiple *N*-glycoprotein encoding genes (see S1 Table) were grouped separately from the remaining genes. Genera were grouped by phylum and sorted by the number of orthologues within each phylum, with phyla abbreviated as follows: C., Crenarchaeota; K., Korarchaeota; M., Micrarchaeota. The underlying source data can be found in S1 Data.

**Table 2. Summary of identified *N*-glycopeptides with noncanonical *N*-glycosite.**

| HVO ID | Name | Description | *N*-glycan type | *N*-glycopeptide sequence | Datasets | PSMs | Predicted processing |
|---|---|---|---|---|---|---|---|
| HVO_0806 | PykA | Pyruvate kinase | Agl15 | IERAGAVD**N**LDEIIQAAHGVMVAR | PXD006877 | 2 | Cyt |
| HVO_2160 | - | Probable secreted glycoprotein | AglB | MPS**N**ANIMGVTPGSR | PXD021874 | 25 | Sec (SPI) |
| HVO_2173 | - | DUF1616 family protein | AglB | LVRGEPASLVLGVG**N**QE | PXD021874 | 2 | ≥ 2 TM |
| HVO_2703 | PanB2 | 3-methyl-2-oxobutanoate hydroxymethyl-transferase | AglB | AHAEAGAFSLVLEHVPA**N**LAK | PXD006877 | 2 | Cyt |
| HVO_A0633 | PilA6 | Pilin PilA | AglB | VVWTSESGS**N**SATLQR | PXD021874 | 20 | Pil (SPIII) |

For each noncanonical *N*-glycopeptide that was identified, the HVO ID, name, and description are given together with the peptide sequence (*N*-glycosite is marked in bold), the *N*-glycan type, the number of corresponding PSMs, the dataset(s) in which it was identified to be *N*-glycosylated, and the predicted processing. Cyt, cytosolic; Pil, type IV pilin pathway; PSM, peptide spectrum match; Sec, Sec pathway; SPI, signal peptidase I; SPIII, signal peptidase III; TM, transmembrane domain.

which was previously suggested to be *N*-glycosylated based on an electrophoretic mobility shift in Δ*aglB* extracts [28], a result that thus far had not been confirmed by MS. Two additional candidates for noncanonical *N*-glycopeptides were identified from the pyruvate kinase PykA (HVO_0806) and the 3-methyl-2-oxobutanoate hydroxymethyltransferase PanB2 (HVO_2703). However, these proteins are predicted to be cytosolic, and the identifications are based on only 2 PSMs each and lack glycan-specific B-ions, since the *m/z*-range of MS2 spectra in the corresponding dataset (PXD006877) does not allow for the identification of mono- and disaccharide B-ions. This renders the identification of noncanonical *N*-glycopeptides in PykA and PanB2 less reliable in comparison to those of HVO_2160 and PilA6 or the identified canonical *N*-glycosites.

Furthermore, it should be noted that *N*-glycans shorter than the AglB-dependent tetra-/pentasaccharide and Agl15-dependent tetrasaccharide were identified. While this is in line with previous reports of shorter *N*-glycans linked to SLG [21], it is unclear whether these represent technical artifacts (e.g., from sample preparation or in-source fragmentation) or biological products (e.g., from *N*-glycan trimming or low substrate specificity of AglB). Therefore, these *N*-glycopeptides have not been further analyzed in this study.

## Identification of likely *O*-glycoproteins

Knowledge about archaeal *O*-glycosylation is so far limited to the early description of disaccharides *O*-linked to the SLG in *Hbt. salinarum* and *Hfx. volcanii* [5,6]. In order to shine new light onto the *O*-glycoproteome of *Hfx. volcanii*, we included the search for potential di-hexose modifications of Ser and Thr in our analysis. While we used the same filtering approach as applied to the identification of *N*-glycopeptides, the shorter length of *O*-glycans reduces the possibilities of detecting *O*-glycan-derived fragment ions. Nevertheless, we identified 23 *O*-glycopeptides corresponding to 20 *O*-glycoproteins (Table 3). Interestingly, the predictions of cellular localisation indicated that both cytosolic and secreted proteins can be *O*-glycosylated in *Hfx. volcanii*. The *O*-glycosylation pathway(s) in archaea remains to be elucidated, hence preventing us from using a deletion mutant lacking *O*-glycans for the stringent evaluation of potential false positives. Nevertheless, the identified *O*-glycopeptides are promising candidates.

## Discussion

Protein glycosylation is involved in a variety of cellular processes in prokaryotes. In this study, we extended the phenotypic analysis of *N*-glycosylation pathway mutants in the model

**Table 3. Summary of identified *O*-glycoprotein candidates.**

| HVO ID | Name | Description | *O*-glyco-peptides | Datasets | PSMs | Predicted processing |
|---|---|---|---|---|---|---|
| HVO_0154 | - | Conserved hypothetical protein | 3 | PXD021874 | 20 | Tat (lipobox) |
| HVO_0306 | - | Probable transmembrane glycoprotein/HTH domain protein | 1 | PXD021874 | 13 | Sec (SPI) |
| HVO_0349 | RpoA1 | DNA-directed RNA polymerase subunit A' | 1 | PXD006877 | 3 | Cyt |
| HVO_0359 | Tef1a1 | Translation elongation factor aEF-1 alpha/peptide chain release factor aRF-3 | 1 | PXD006877 | 7 | Cyt |
| HVO_0654 | Rpl43e | 50S ribosomal protein L43e | 1 | PXD006877 | 4 | Cyt |
| HVO_0677 | AspS | Aspartate–tRNA(Asp/Asn) ligase | 1 | PXD011056 | 6 | Cyt |
| HVO_0778 | Ths3 | Thermosome subunit 3 | 1* | PXD013046 | 8 | Cyt |
| HVO_0869 | GltB | Glutamate synthase (ferredoxin) large subunit | 1 | PXD006877 | 2 | Cyt |
| HVO_1148 | Rps15 | 30S ribosomal protein S15 | 1 | PXD006877 | 4 | Cyt |
| HVO_1198 | - | UspA domain protein | 1 | PXD006877 | 8 | Cyt |
| HVO_1597 | - | Conserved hypothetical protein | 1 | PXD021874 | 4 | Tat (lipobox) |
| HVO_2071 | - | Probable secreted glycoprotein | 2* | PXD021874 | 12 | Sec (SPI) |
| HVO_2160 | - | Probable secreted glycoprotein | 1* | PXD021874 | 5 | Sec (SPI), ArtA |
| HVO_2172 | - | Conserved hypothetical protein | 1* | PXD021874 | 3 | Sec (SPI) |
| HVO_2226 | TrpD2 | Probable phosphoribosyltransferase (homolog to anthranilate phosphoribosyltransferase) | 1 | PXD006877 | 6 | Cyt |
| HVO_2413 | Tef1a2 | Translation elongation factor aEF-1 alpha/peptide chain release factor aRF-3 | 1 | PXD006877 | 2 | Cyt |
| HVO_2487 | Asd | Aspartate-semialdehyde dehydrogenase | 1 | PXD006877 | 3 | Cyt |
| HVO_2580 | NadB | L-aspartate oxidase | 1 | PXD006877 | 3 | Sec (SPI) |
| HVO_A0380 | DppA8 | ABC-type transport system periplasmic substrate-binding protein (probable substrate dipeptide/oligopeptide) | 1 | PXD021874 | 2 | Tat (lipobox) |
| HVO_B0050 | CobN | ATP-dependent cobaltochelatase subunit CobN | 1 | PXD006877 | 2 | Cyt |

For each protein that has been identified to be likely *O*-glycosylated in this study, the HVO ID, name and description are given together with the number of identified *O*-glycopeptides, the number of corresponding PSMs and dataset(s), and the predicted processing.

ArtA, archaeosortase A substrate; Cyt, cytosolic; HTH, helix–turn–helix; lipobox, conserved cleavage site motif for lipoproteins; PSM, peptide spectrum match; Sec, Sec pathway; SPI, signal peptidase I; Tat, twin arginine translocation pathway.

archaeon *Hfx. volcanii*. Our results revealed that *N*-glycosylation is not only involved in cell growth and motility, but also affects colony morphology and cell shape. Furthermore, we could show that the deletion of *agl15*, which is required for an *N*-glycosylation type that was so far described to occur only under low-salt conditions, has phenotypic effects under optimal salt conditions as well.

In order to gain insights into the underlying molecular basis for these effects, we developed a stringent workflow for the in-depth glycoproteomic analysis of *Hfx. volcanii*. With the identification of 45 *N*-glycoproteins, this approach resulted in the most extensive archaeal *N*-glycoproteome described so far. Importantly, this glycoproteomic workflow is applicable to a broad range of archaea with known *N*-glycosylation pathways and even allows for the reanalysis of existing MS data as shown by the incorporation of ArcPP datasets. Furthermore, it revealed the surprising concurrence of the AglB- and Agl15-dependent *N*-glycosylation pathways. The identification of *N*-glycopeptides corresponding to both pathways implies their activity under the same optimal salt conditions. Remarkably, datasets in which AglB- and Agl15-dependent *N*-glycopeptides were identified comprised a broad range of growth conditions (mid-logarithmic and early-stationary growth phase, planktonic and biofilm cells, and oxidative stress conditions). While quantitative analyses are beyond the scope of this manuscript, these results indicate that neither AglB- nor Agl15-dependent *N*-glycosylation is limited to a specific

growth condition. Additionally, our qualitative analysis showed that the 2 independent *N*-glycosylation pathways are able to modify the same *N*-glycosites.

It should be noted that Agl15-dependent *N*-glycans were not identified in samples from the Δ*aglB* mutant. Similar band intensities for Pro-Q Emerald 300 stained glycoproteins in an LDS-PAGE comparing WT, Δ*aglB*, and Δ*agl15* samples are therefore most likely related to a different type of glycosylation. This could include *O*-glycosylation, which was detected in samples of all 3 analyzed strains, and which is a known modification of SLG [6], as well as a third *N*-glycosylation type for which the corresponding biosynthesis pathway has not been identified so far [24]. Nevertheless, the lack of Agl15-dependent *N*-glycans was surprising because previous studies indicated that this type of glycosylation increases in the absence of AglB-dependent glycosylation [36]. However, these studies were performed with purified SLG, allowing for a higher sensitivity than the cellular fractions analyzed here. The decreased stability of the SLG in the Δ*aglB* mutant might have further hampered the identification of Agl15-dependent *N*-glycopeptides in the Δ*aglB* mutant. Furthermore, the *N*-glycosite harboring Agl15-dependent glycans analyzed by Kaminski and colleagues (N498) differs from those identified here (N274 and N279). Therefore, it cannot be excluded that changes in the abundance of Agl15-dependent *N*-glycopeptides depend on the *N*-glycosite. It is worth noting that even in the dataset PXD011056, which contains samples from cultures grown under low-salt conditions, no increase in the identification of Agl15-dependent *N*-glycopeptides could be seen. In line with this, the vast majority of enzymes corresponding to the Agl15-dependent *N*-glycosylation pathway are less abundant under low-salt conditions compared to optimal salt conditions, as shown by the quantification performed by Jevtić and colleagues (PXD011056, [44]). While different low-salt regimes were applied in the various studies, it can be concluded that Agl15-dependent *N*-glycosylation is not specific to low-salt conditions. This is in line with the physiological phenotypes observed here for the Δ*agl15* mutant under optimal salt conditions.

Overall, our results highlight a remarkable complexity of the glycosylation pathways in *Hfx. volcanii*, which poses questions about their regulation. Interestingly, AglB itself was found to be *N*-glycosylated. The eukaryotic counterpart of AglB, the OST complex subunit STT3, has previously been shown to be *N*-glycosylated as well, with at least 1 *N*-glycan playing an essential role in the assembly and function of the complex [45–47]. Despite the evolutionary proximity of archaeal AglB and eukaryotic STT3 [48], the *N*-glycosylation of AglB itself had not been previously revealed. While the archaeal OST acts as a single-subunit enzyme instead of a complex, its *N*-glycan could nevertheless serve regulatory functions and/or interact with other proteins.

The importance of *N*-glycosylation for S-layer stability, motility, and adhesion via the modification of SLG, archaellins and type IV pilins, respectively, has been shown previously [26–28]. However, the extent of *N*-glycosylation in *Hfx. volcanii* revealed here indicates that additional proteins could be involved in these processes and that a variety of other cellular functions may be affected. For example, while *N*-glycosylation has been shown to be crucial for mating [29], SLG has been suggested to play a role in this process mainly because it is the sole component of the *Hfx. volcanii* S-layer and is known to be highly *N*-glycosylated. However, our results show that a multitude of cell surface proteins in *Hfx. volcanii* are *N*-glycosylated, some of which have been linked to mating. A recent transcriptomic study of mating in *Hfx. volcanii* included 8 genes, for which we showed *N*-glycosylation of their gene products, in a cluster that indicated elevated transcript levels during the initial time points of mating [49]. Besides SLG, AglB, and CoxB1, this comprised various proteins of unknown function (HVO_1749, HVO_1981, HVO_2071, HVO_2533, and HVO_C0054). Interestingly, HVO_2533 is also a predicted ArtA substrate, and *artA* deletion strains have been shown to be

mating deficient [50]. Other confirmed or predicted ArtA substrates that are *N*-glycosylated are SLG and HVO_2160, the largest *Hfx. volcanii* protein.

The lack of ArtA has also been associated with a defect in transitioning from rod- to disk-shaped cells [51,52]. Since our results revealed shape phenotypes of Δ*aglB* as well as Δ*agl15*, the *N*-glycosylation of ArtA substrates may be important for their function and could therefore be involved in the regulation of *Hfx. volcanii* cell shape. SLG, as the most abundant ArtA substrate and sole component of the cell wall, is likely to be involved in cell shape determination of *Hfx. volcanii*. Its *N*-glycosylation has been shown to be required for the proper assembly of the S-layer [26,29,53]. Our results indicate that while cells lacking AglB-dependent *N*-glycosylation were still able to form WT-like disks, their ability to form rods was impaired. Considering that populations of SLG peptides harboring either AglB- or Agl15-dependent *N*-glycans were identified, but no molecules that were modified by both *N*-glycosylation pathways, it is tempting to speculate that spatiotemporal changes in the *N*-glycosylation of SLG could be involved in the cell shape regulation of *Hfx. volcanii*. Furthermore, while conformational changes of SLG have been revealed in response to different environmental conditions [54], the conformation of differentially *N*-glycosylated SLG remains to be elucidated.

In addition to SLG, the AglB-dependent *N*-glycosylation of ABC transporters that we identified here could have an effect on (micro-)nutrient acquisition in *Hfx. volcanii*, which has been linked to cell shape as well [31]. This may also explain differing growth curves between the WT and especially the Δ*aglB* strain. Finally, the *N*-glycosylation of SecD, a component of the secretory pathway, could affect cell surface biogenesis and thereby cell shape as well as other surface-related processes.

In conclusion, the extensive and complex *N*-glycoproteome of *Hfx. volcanii* that we revealed here shines new light on a variety of cellular functions. While the specific effects of *N*-glycosylation on the various identified proteins remain to be studied in detail, their identification represents an essential first step in analyzing the roles of *N*-glycosylation in archaea. The importance of this is highlighted not only by the various phenotypes of *N*-glycosylation pathway mutants but also by the phylogenetic conservation of identified *N*-glycoproteins with unknown functions within Halobacteria.

## Materials and methods

### Strains and growth conditions

*Hfx. volcanii* H53, as well as the Δ*aglB* and Δ*agl15* mutants, was grown at 45°C in liquid or on solid agar semi-defined casamino acid (Hv-Cab) medium [31]. H53 cultures were supplemented with tryptophan (+Trp) and uracil (+Ura) at a final concentration of 50 μg ml$^{-1}$. The generation of the Δ*aglB* and Δ*agl15* strains from the background strain H53 (Δ*pyrE2*; Δ*trpA*) was described previously [36,55], and both strains were obtained from the Eichler laboratory. Each mutant carried a *trpA* insertion in the respective gene and was therefore grown on Hv-Cab +Ura.

For direct comparison of colony morphology and motility phenotypes, all strains were grown on Hv-Cab +Trp +Ura, but phenotypes of deletion mutants were confirmed on Hv-Cab +Ura. Colony morphology was imaged after 5 days of incubation at 45°C using a Nikon D3500 DX.

### Motility assay

Motility assays were performed as described previously [30] using Hv-Cab media containing 0.35% agar supplemented with Ura and Trp as required for the analyzed strain. Halos around

the stab-inoculation site were imaged after 5 days of incubation at 45˚C using a Nikon D3500 DX.

## Growth curve

Liquid Hv-Cab starting cultures (5 mL) were inoculated from single colonies and grown at 45˚C under shaking (250 rpm). After reaching an $OD_{600}$ of 0.3 to 0.5 (measured in the culture tube, path length approximately 1.5 cm), all cultures were diluted to an $OD_{600}$ of 0.05 in 20 mL to obtain comparable conditions at the start of the growth curve. Subsequently, $OD_{600}$ was monitored for 74 hours; these measurements were performed by pipetting 250 μL of culture into a 96-well plate and measuring the absorption with a Biotek PowerWaveX2 microplate spectrophotometer. When cultures reached an OD above 0.75, samples were diluted 1:5 before measurement.

## Cell shape assessment

Liquid Hv-Cab (5 mL) was inoculated with single colonies and grown at 45˚C under shaking (250 rpm). At early-logarithmic ($OD_{600}$ of 0.03 to 0.05), mid-logarithmic ($OD_{600}$ of 0.35 to 0.5), and late-logarithmic ($OD_{600}$ of 1.6 to 2.0) growth phase, 1 mL of culture was centrifuged (5 minutes, 6,000 g) and resuspended in 20 μL of medium (for early-logarithmic phase, correspondingly higher volumes were used for higher ODs). Resuspended cells were observed on a Leica DMi8 microscope at 100× magnification using differential inference microscopy (DIC). Images were taken with a Leica DFC9000GT camera attached to the microscope.

For quantitative cell shape analyses, images were taken in brightfield instead of DIC mode and processed with CellProfiler (version 4.1.3) [34]. Cells were identified using the IdentifyPrimaryObjects and IdentifySecondaryObjects (propagation method) functions with a typical object diameter of 15 to 50 pixels, robust background thresholding, and a lower threshold bound of 0.83. After combining primary and secondary objects and merging touching objects, the MeasureObjectSizeShape function was employed to determine the eccentricity of cells, with 0 corresponding to a circle and 1 corresponding to a line shape. For each strain and biological replicate, 2 representative images were analyzed, corresponding to 300 to 750 cells for strain and condition. For each condition and strain combination, a *t* test was performed, testing for equal means and assuming unequal variances. Resulting *p*-values were corrected for multiple testing using the Benjamini–Hochberg method.

## Cell fractionation

A 5-mL Hv-Cab starting culture (inoculated from single colonies for biological replicates) was grown at 45˚C under shaking (250 rpm) until it reached an $OD_{600}$ of 0.3 to 0.5 (measured in the culture tube, path length approximately 1.5 cm). The culture was then diluted to 25 mL ($OD_{600}$ of 0.01) and was kept growing under the same conditions. Once the culture reached an $OD_{600}$ of 0.3, a 10-mL sample was taken, while the remaining culture continued growing for another 24 hours, reaching early stationary phase ($OD_{600} > 1.7$), at which point a second 10-mL sample was taken. Each sample was processed as follows immediately after it has been taken.

i. The culture was centrifuged at 6,000 g for 10 minutes to separate the cells from the SN. The SN fraction was transferred into new tubes and centrifuged again 2 more times, once at 10,000 g for 10 minutes and once at 16,000 g for 20 minutes, to remove contaminating cells and potential large cell debris. Afterwards, the SNs from both samples (mid-logarithmic and early-stationary growth phase) were combined and concentrated to approximately 150 μL

in centrifugal filter units with a 3-kDa molecular mass cutoff (Amicon Ultra Centrifugal Filters, 0.5 ml, Millipore, centrifugation at 14,000 g).

ii. The cell pellets from step (i) of both samples were resuspended and combined in 1-mL PBS (2.14 M NaCl, 2.68 mM KCl, 10.14 mM $Na_2HPO_4$, 1.76 mM $KH_2PO_4$, pH 7.4) containing 10 mM EDTA as well as 1 mM 4-(2-aminoethyl) benzenesulfonyl fluoride hydrochloride and 1 mM phenylmethylsulfonyl fluoride as protease inhibitors. Cells were lysed by freezing (−80˚C) and thawing (on ice) them 3 times. Cellular DNA was digested by adding 10 μg of DNAse I and incubating the mixture at 37˚C for 30 minutes. The lysate was centrifuged at 10,000 g for 5 minutes at 4˚C to pellet unlysed cells. Subsequently, the SN was centrifuged for 30 minutes at 300,000 g and 4˚C in a Beckman TL-100 ultracentrifuge. The SN of this step was transferred to a new tube and centrifuged again (using the same settings) to remove potential membrane contaminations. The resulting SN corresponds to the Cyt fraction.

iii. The pellet after the first ultracentrifugation step was washed in 600 μL of ice-cold PBS and centrifuged again using the same settings. After carefully removing the SN, the pellet representing the Mem fraction was resuspended in 300 μL 100 mM Tris/HCl buffer (pH 7.4) containing 2% SDS.

Each cellular fraction was transferred to centrifugal filter units with a 3-kDa molecular mass cutoff, washed 3 times with 400 μL $H_2O$, followed by a protein solubilization step with 400 μL 100 mM Tris/HCl buffer (pH 7.4) containing 2% SDS, incubation at 55˚C for 15 minutes, and 2 additional washes with 400 μL $H_2O$. Afterwards, the protein concentration of each fraction was determined using a bicinchoninic acid assay (BCA Protein Assay Kit by Thermo Fisher Scientific). Samples were stored at −80˚C until further use.

## SDS-PAGE and glycoprotein staining

Protein samples of each fraction (7.5 μg) were mixed with NuPAGE loading buffer and dithiothreitol (DTT, final concentration 50 mM), incubated at 55˚C for 15 minutes and then separated by electrophoresis on a 4% to 12% v/v Bis-Tris NuPAGE gel (1.5 mm, 10 well) using MOPS SDS running buffer (NuPAGE). Two gels were run in parallel, one of which was stained overnight with Coomassie brilliant blue for protein detection (destaining with $H_2O$), while the other was stained for glycoproteins using the Pro-Q Emerald 300 glycoprotein staining kit (Invitrogen) following the manufacturer's instructions. SeeBlue Plus2 prestained protein marker (Thermo Fisher Scientific) was used for Coomassie-stained gels, whereas CandyCane glycoprotein marker (Invitrogen) was used for Pro-Q Emerald 300 stained gels. In addition, BSA (5 μg, Thermo Fisher Scientific) was loaded as a control, which is not modified by *N*-glycans but is stained by periodic acid–Schiff staining [56], potentially due to *O*-glycosylation. Furthermore, CcmG from *Rhodobacter capsulatus*, recombinantly expressed and purified from *Escherichia coli* [57], was loaded (15 μg, generously provided by Fevzi Daldal and Andreia Verissimo) as a non-glycosylated control. Gels were imaged using an Amersham Imager 600 (GE Healthcare Life Sciences). UV light was used to image Pro-Q Emerald 300 stained gels, and exposure times were adjusted to 0.1 seconds to result in minimal signal from the non-glycosylated control.

## Mass spectrometric analysis

Protein samples of each fraction (30 μg for Mem and Cyt and 10 μg for SN) were processed and digested using S-Trap mini spin columns (ProtiFi) and following the manufacturer's instructions. Samples were mixed with an equal volume of solubilization buffer (10% SDS in

100 mM triethylammonium bicarbonate (TEAB), pH 7.55) and subsequently reduced by adding 20 mM DTT (15 minutes, 55°C) and alkylated by adding 40 mM iodoacetamide (30 minutes, room temperature, dark). After acidifying the solution with phosphoric acid (final concentration 1.2%), S-Trap binding buffer (90% methanol, 100 mM TEAB, pH 7.1) was added at a ratio of 1:7 (sample volume: binding buffer volume). The mixture was loaded onto the S-Trap spin column and centrifuged at 4,000 g for 30 seconds. Captured proteins were washed 3 times with 400 μL S-Trap binding buffer. Proteases were added at a ratio of 1:25 (enzyme weight: sample protein weight) in 125 μL digestion buffer (50 mM TEAB for trypsin; 10 mM phosphate buffer, pH 8.0 for GluC). Digestion was performed overnight at 37°C. Peptides were eluted in 3 steps: 80 μL $H_2O$, followed by 80 μL 0.2% formic acid, and finally 80 μL 50% acetonitrile (ACN). The spin column was centrifuged at 1,000 g for 60 seconds at each step. Eluted peptides from the different steps were pooled and dried in a vacuum centrifuge.

All samples were desalted using homemade C18 stage-tips (3 M Empore Discs) as previously described [70], with minor modifications. Briefly, columns were conditioned with 100 μL ACN and equilibrated 2 times with 100 μL 0.1% trifluoroacetic acid (TFA). Samples were resuspended in 0.1% TFA and loaded onto the stage-tip. After washing 3 times with 100 μL 0.1% TFA, peptides were eluted with 2 times 80 μL 0.1% formic acid in 60% ACN and dried in a vacuum centrifuge.

Peptides were reconstituted in 0.1% formic acid (solvent A) and analyzed with a Dionex Ultimate 3000 UPLC (Thermo Fisher Scientific) coupled via a nanospray source to a Q-Exactive HFX tandem MS (Thermo Fisher Scientific) operated in positive ion mode. Approximately 1 μg of peptides were desalted on a trap column (Acclaim PepMap100 C18, 5 μm, 100 Å, 300 μm i.d. × 5 mm, Thermo Fisher Scientific) and then separated on an in-house packed column (75 μm i.d. × 25 cm fused silica capillary packed with 3 μm ReproSil-Pur C18 beads (Dr. Maisch)). A 60-minute gradient was used with a flow rate of 400 μL $mL^{-1}$. Solvent B (0.1% formic acid in 80% ACN) was increased from 4% to 21% over 40 minutes, followed by an increase to 40% over 20 minutes. Afterwards, to wash and re-equilibrate the column, solvent B was increased to 90% over 2 minutes, kept constant at 90% for 6 minutes, and then decreased to 4% over 2 minutes with subsequent constant flow at 4% for at least 10 minutes.

Data-dependent acquisition was performed, acquiring MS1 spectra over a range of 400 to 2,000 m/z with a resolution of 60,000, an AGC target of 1e6, and a maximum injection time of 100 ms. The top 20 peaks were selected for HCD fragmentation with stepped normalized collision energy (25, 30, and 35 NCE). MS2 spectra were acquired with a fixed first mass (130 m/z), a resolution of 15,000, an AGC target of 1e5, and a maximum injection time of 200 ms. A dynamic exclusion list (10 seconds) was used, and charge states 1 and >6 were excluded. MS raw files generated for this study have been uploaded to the ProteomeXchange Consortium via the PRIDE partner repository [58] with the data set identifier PXD021874.

## Bioinformatic analysis of MS data

The analysis of MS data followed the general workflow established by the ArcPP [23] using the Python framework Ursgal [59], with modifications as follows. Briefly, MS raw files were converted into mzML files using ThermoRawFileParser [60], followed by conversion into MGF format by pymzML [61]. Protein database searches were performed against the ArcPP *Hfx. volcanii* theoretical proteome (4,074 proteins, version 190606, June 6, 2019, https://doi.org/10.5281/zenodo.3565631), supplemented with common contaminants, and decoys using shuffled peptide sequences. MSFragger (version 2.3) [62], X!Tandem (version vengeance) [63], and MS-GF+ (version 2019.07.03) [64] were employed as search engines. Search parameters for the reanalysis of ArcPP datasets matched those described previously [23]. For the dataset

**Table 4. Glycans included as potential modifications in protein database searches.**

| Amino acid | Glycan composition | Chemical composition | Unimod ID |
|---|---|---|---|
| N | Hex(1) | $C_6H_{10}O_5$ | 41 |
| N | Hex(1)HexA(1) | $C_6H_8O_6$ | 1427 |
| N | Hex(1)HexA(2) | $C_{18}H_{26}O_{17}$ | - |
| N | Hex(1)HexA(2)MeHexA(1) | $C_{25}H_{36}O_{23}$ | - |
| N | Hex(2)HexA(2)MeHexA(1) | $C_{31}H_{46}O_{28}$ | - |
| N | SO3Hex(1) | $C_6H_{10}O_8S_1$ | - |
| N | SO3Hex(1)Hex(1) | $C_{12}H_{20}O_{13}S_1$ | - |
| N | SO3Hex(1)Hex(2) | $C_{18}H_{30}O_{18}S_1$ | - |
| N | SO3Hex(1)Hex(2)dHex(1) | $C_{24}H_{40}O_{22}S_1$ | - |
| S/T | Hex(2) | $C_{12}H_{20}O_{10}$ | 512 |

Given are the glycan composition (dHex, deoxyhexose; Hex, hexose; HexA, hexuronic acid; MeHexA, methyl-hexuronic acid; SO3Hex, sulfated hexose), the chemical composition, the Unimod ID, and the amino acid that has been specified as potential site of modification.

PXD021874 (generated for this study), a precursor mass tolerance of +/−10 ppm and a fragment mass tolerance of 10 ppm were used, 2 and 3 missed cleavages were allowed for data derived from samples digested with Trypsin and GluC, respectively, and oxidation of M, N-terminal acetylation were included as potential modifications, while carbamidomethylation of C was required as fixed modification. In addition, for all datasets, all known compositions of AglB- and Agl15-dependent *N*-glycans have been added as potential modifications of N (Table 4). Hex(2) was included as a potential modification of S or T for the search of *O*-glyco-peptides. Since most protein database search engines do not support the search for multiple different modifications of the same amino acid, each *N*-glycan composition was searched for separately. Unfortunately, this means that we could not assess whether peptides containing 2 *N*-glycosites could be modified with 1 AglB- and 1 Agl15-dependent glycan.

Results from all cellular fractions of one sample were merged before statistical post-processing with Percolator (version 3.4) [65]. Subsequently, results from different search engines and from all searches for different glycan compositions were combined using the combined PEP approach as described before [23,59]. After removal of PSMs with a combined PEP > 1%, results were sanitized, i.e. for spectra with multiple differing PSMs, only the PSM with the best combined PEP was accepted. Peptide and protein FDRs were calculated as described previously [23], while treating each peptidoform (peptide + modification) separately and only peptidoforms with an FDR ≤1% and proteins with an FDR ≤0.5% were accepted.

Spectra corresponding to glycopeptide spectrum matches were searched for glycopeptide-specific fragment ions using the glycopeptide_fragmentor node in Ursgal using the glycopeptide ion matching functionality of SugarPy [66]. The fragment ion mass tolerance was set to 20 ppm, and at least 1 B-ion and 2 Y-ions were required to be found in at least 1 spectrum of each glycopeptide. Finally, glycopeptides were required to be identified in at least 2 replicates (as defined for each dataset).

### Prediction of protein subcellular localization and protein domains

Secretion signals and the probable subcellular localization were determined as previously described [23]. Candidates for ArtA processing were taken from [67]. Further protein domains were predicted with InterPro [68], and arCOGlet categories for each protein were obtained based on the latest release (December 2015) [69].

## Phylogenetic analyses

For analysis of phylogenetic distributions, we used OrthoDB v10.1 [42]. The locus tag (e.g., HVO_2072 for SLG) was used to interrogate OrthoDB, and groups were extracted at the taxonomic levels Halobacteria, Euryarchaeota, and Archaea (October 2020). At each level, OrthoDB lists the number of species having an ortholog and the number of species analyzed at that level. For each group and level, a FASTA file is provided, and the FASTA files at levels Halobacteria and Archaea were downloaded for a more detailed, script-based analysis. At the level Halobacteria, the analysis was performed at the species level. Four species were represented by 2 strains, and only one of them was considered, leaving a total of 161 species at the level Halobacteria. For the domain Archaea, the analysis was performed at the genus level. OrthoDB codes consist of a species identifier (the NCBI TaxID with a strain serial number) and a protein identifier. For transformation of TaxID to species/genus, the NCBI taxonomy was inquired (https://www.ncbi.nlm.nih.gov/taxonomy) (accessed October 22, 2020).

For gene synteny analysis, the SyntTax server was used [41] (accessed October 2020). At the time of usage, the server analyzed 94 genomes from the taxonomic class Halobacteria. It should be noted that only the main chromosomes are analyzed within the SyntTax server.

## Supporting information

**S1 Fig. Proteins from cellular fractions of WT, Δ*agl15*, and Δ*aglB* exhibit comparable glycoprotein staining.** Samples from WT, Δ*agl15*, and Δ*aglB* strains grown to mid-logarithmic and early-stationary growth phase (mixed using equal culture volumes) were fractionated into Mem, Cyt, and culture SN. Equal protein amounts for each strain (7.5 μg) were separated by LDS-PAGE and stained using Coomassie brilliant blue (**A**) or Pro-Q Emerald 300 glycoprotein staining (**B**). As controls, BSA (5 μg), which does not exhibit *N*-glycosylation but shows staining by periodic acid–Schiff staining [56], and the *Rhodobacter capsulatus* CcmG (15 μg) recombinantly expressed and purified from *Escherichia coli*, representing a non-glycosylated protein, were used. UV light exposure times have been adjusted to 0.1 seconds to result in minimal signal from the non-glycosylated control. The band corresponding to SLG has been marked (arrow), as its electrophoretic mobility is well established [51,52]. The images are representative for 2 biological replicates. Different staining procedures were performed on separate gels using the same biological replicates. The original images can be found in S1 Raw Images. Cyt, cytosol; Mem, membrane; SLG, S-layer glycoprotein; SN, supernatant; WT, wild-type.
(PDF)

**S2 Fig. MS2 spectra of SLG *N*-glycosites N274 and N279 strongly support modification by AglB- as well as Agl15-dependent *N*-glycans.** Annotated spectra for SLG *N*-glycopeptides comprising the peptide sequence VGIANSSATNTSGSSTGPTVE with AglB- (**A**) and Agl15-dependent (**B**) *N*-glycans attached to the *N*-glycosites N274 and N279. Measured raw peaks are shown in gray, annotated a- and b-ions in purple, y-ions in yellow, and *N*-glycopeptide-specific Y- and B-ions in cyan. Insets illustrate the peptide sequence coverage through a- or b-ions (purple) and y-ions (yellow) (in both cases detected ions shown as wide bar, missing ions shown as line), as well as the coverage of Y- and B-ions (detected ions shown in cyan). The underlying source data for A and B can be found in S1 Data. MS2, tandem mass spectrum; SLG, S-layer glycoprotein.
(PDF)

**S3 Fig. More proteins and peptides have been identified in PXD021874 than in any other dataset of the ArcPP.** The number of identified peptides (light gray) and proteins (dark gray)

for each dataset is shown as a barplot (sorted by the total number of identified proteins). The underlying source data can be found in S1 Data. ArcPP, Archaeal Proteome Project.
(PDF)

**S4 Fig. Agl15-dependent *N*-glycosylation of Ths3 is strongly supported by MS2 fragment ion series.** Shown is an annotated spectrum for the Ths3 *N*-glycopeptide with sequence KSEVDTEYNITSVDQLTAAIDAEDSELR harboring an Agl15-dependent *N*-glycan. Measured raw peaks are shown in gray, annotated a- and b-ions in purple, y-ions in yellow, and *N*-glycopeptide-specific Y- and B-ions in cyan. Insets illustrate the peptide sequence coverage through a- or b-ions (purple) and y-ions (yellow) (in both cases detected ions shown as wide bar, missing ions shown as line), as well as the coverage of Y- and B-ions (detected ions shown in cyan). The underlying source data can be found in S1 Data. MS2, tandem mass spectrum; Ths3, thermosome subunit 3.
(PDF)

**S5 Fig. Distribution of *N*-glycoproteins across Halobacteria.** For each identified *N*-glycoprotein (x-dimension), the percentage of species within the class of Halobacteria that carry an orthologue is presented. Furthermore, for each species (y-dimension), the percentage of *N*-glycoproteins for which an orthologue is present within the species' genome is shown. Percentages are shown as heat, ranging from yellow (100%) to purple (0%). The underlying source data can be found in S1 Data.
(PDF)

**S6 Fig. *N*-glycopeptides with noncanonical *N*-glycosites are strongly supported by MS2 fragment ion series.** Annotated spectra for noncanonical *N*-glycosites within the peptide sequences VVWTSESGSNSATLQR (**A**) and MPSNANIMGVTPGSR (**B**) corresponding to the protein pilA6 and HVO_2160, respectively, are shown. Both *N*-glycopeptides are modified by an AglB-dependent glycan, and the site of attachment is indicated. Measured raw peaks are shown in gray, annotated a- and b-ions in purple, y-ions in yellow, and *N*-glycopeptide-specific Y- and B-ions in cyan. Insets illustrate the peptide sequence coverage through a- or b-ions (purple) and y-ions (yellow) (in both cases detected ions shown as wide bar, missing ions shown as line), as well as the coverage of Y- and B-ions (detected ions shown in cyan). The underlying source data for A and B can be found in S1 Data. MS2, tandem mass spectrum.
(PDF)

**S1 Table. Analysis of phylogeny, gene synteny, and protein domains of identified *N*-glycoproteins.** For each protein that was identified to be *N*-glycosylated, the HVO ID, description, InterPro domains and arCOGlet classification are given together with results of phylogenetic and gene synteny analysis for the genes encoding the identified *N*-glycoproteins.
(PDF)

**S1 Raw images. Uncropped images of Coomassie and Pro-Q Emerald 300 stained gels from S1 Fig.** Samples from WT, Δ*agl15*, and Δ*aglB* strains grown to mid-logarithmic and early-stationary growth phase (mixed using equal culture volumes) were fractionated into Mem, Cyt and culture SN. Equal protein amounts for each strain (7.5 μg) were separated by LDS-PAGE and stained using Coomassie brilliant blue (page 1) or Pro-Q Emerald 300 glycoprotein staining (page 2). As controls, BSA (5 μg), which does not exhibit *N*-glycosylation but shows staining by periodic acid–Schiff staining [56], and the *Rhodobacter capsulatus* CcmG (15 μg) recombinantly expressed and purified from *Escherichia coli*, representing a non-glycosylated protein, were used. UV light exposure times have been adjusted to 0.1 seconds to result in minimal signal from the non-glycosylated control. Cyt, cytosol; Mem, membrane; SN,

supernatant; WT, wild-type.
(PDF)

**S1 Data. Underlying source data for Figs 2A, 2C, 4A, 4B and 5 and S2A, S2B, S3, S4, S5, S6A and S6B Figs.** For each figure panel, a separate sheet with the underlying source data is provided.
(XLSX)

## Acknowledgments

We would like to thank the Pohlschroder and Garcia lab members for helpful discussions and their support in MS data acquisition. Furthermore, providing access to the Python peptide fragmentor module by Christian Fufezan is greatly appreciated.

## Author Contributions

**Conceptualization:** Stefan Schulze, Benjamin A. Garcia, Mechthild Pohlschroder.

**Data curation:** Stefan Schulze, Friedhelm Pfeiffer.

**Formal analysis:** Stefan Schulze.

**Funding acquisition:** Stefan Schulze, Benjamin A. Garcia, Mechthild Pohlschroder.

**Investigation:** Stefan Schulze, Friedhelm Pfeiffer.

**Methodology:** Stefan Schulze.

**Project administration:** Mechthild Pohlschroder.

**Resources:** Benjamin A. Garcia.

**Software:** Stefan Schulze.

**Supervision:** Benjamin A. Garcia, Mechthild Pohlschroder.

**Validation:** Stefan Schulze, Friedhelm Pfeiffer.

**Visualization:** Stefan Schulze, Friedhelm Pfeiffer.

**Writing – original draft:** Stefan Schulze.

**Writing – review & editing:** Friedhelm Pfeiffer, Benjamin A. Garcia, Mechthild Pohlschroder.

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
