## [Editor Report · Decision Letter 0]

18 Feb 2021

Dear Dr. Pohlschroder, 

Thank you for submitting your manuscript entitled "Comprehensive glycoproteomics shines new light on the complexity and extent of glycosylation in archaea" for consideration as a Research Article by PLOS Biology.

Your manuscript has now been evaluated by the PLOS Biology editorial staff, as well as by an academic editor with relevant expertise, and I am writing to let you know that we would like to send your submission out for external peer review.

Please re-submit your manuscript within two working days, i.e. by Feb 20 2021 11:59PM.

We will consider this manuscript as a Methods and Resources Article, please choose that option as the Article type where corresponds. 

Kind regards,

Paula

---

Associate Editor

PLOS Biology

---

## [Decision Letter · Decision Letter 1]

19 Mar 2021

Dear Dr. Pohlschroder,

Thank you very much for submitting your manuscript "Comprehensive glycoproteomics shines new light on the complexity and extent of glycosylation in archaea" for consideration as a Research Article at PLOS Biology. Your manuscript has been evaluated by the PLOS Biology editors, an Academic Editor with relevant expertise, and by several independent reviewers.

In light of the reviews (below), we are pleased to offer you the opportunity to address the comments from the reviewers in a revised version that we anticipate should not take you very long. We will then assess your revised manuscript and your response to the reviewers' comments and we may consult the reviewers again.

We expect to receive your revised manuscript within 2 months.

In particular, reviewer #1 has some questions about the glycosylation of the thermosome subunit 3 and gives some alternative hypothesis, and says that you should’ve shown the fraction of Haloferax species/strains that encode those proteins. Reviewer #2 suggests to cut out a section about the observation that the agl15 mutant grows at higher ODs or explain the implications in the discussion. This reviewer also says that you have to show measurements to show difference in cell morphology, asks for clarifications, says that if there was a mixed population of different modified SLG molecules, you should discuss about whether it points out to a spatiotemporal regulation within individual cells, or different profiles during shape transitions throughout growth phases, and asks for more information about the strains used.

**IMPORTANT - SUBMITTING YOUR REVISION**

*Resubmission Checklist*

*Published Peer Review*

*PLOS Data Policy*

*Blot and Gel Data Policy*

Sincerely,

Paula 

---

Associate Editor,

pjaureguionieva@plos.org,

PLOS Biology

REVIEWS:

Reviewer #1: Archaeal glycosylation.

Reviewer #2: Cellular Organization and Behavior in Archaea.

Reviewer #1: General comments

The manuscript of Schulze et al. represent a long-overdue comprehensive analysis of the glycoproteome of the model archaeon Haloferax volcanii. It reveals interesting overlaps in glycosylation between the major (AglB-dependent) and the minor (Agk15-dependent) N-glycosylation pathways, and also hints at O-glycosylations. When combined with previous ArcPP data from these authors, there are 45 high-confidence glycoproteins, many of which are of unknown function. Overall, the methodology used is conservative, and therefore highly unlikely to erroneously identify a non-existing glycosylation, but the authors should acknowledge that for low abundance proteins glycosylation could be missed due to the stringency and those 45 probably represent a lower bound, especially since some membrane-associated proteins may be only highly expressed under specific conditions. The phenotypic characterization is convincing, but does not add much to what is already known about glycosylation mutants.

Specific comments

Thermosome subunit 3 is the only exclusively Agl15-dependent glycoprotein, the only cytosolic glycoprotein and is also o-glycosylated! The authors speculate that some of it may be surface associated (maybe as a moonlighting role that is not thermosome related) but I could not find indication as to whether the glycosylated peptides were exclusively primarily in the membrane and secreted fractions. I think that these 3 observations are unlikely to be coincidental. My null hypothesis (which may be wrong, but should at least be considered) is that being part of a chaperone complex implies that this could just be a protein that gets glycosylated at random because it is highly abundant and interacts closely with many proteins, including, presumably, glycosyl-transferases. I understand that the data from all fractions were pooled, but once high-confidence glycosylated peptides were detected one can go back to the individual protein-associated peptides and glycopeptides and see how many peptides are identified in the cytosolic vs. the membrane fractions. This is by no means truly quantitative, but should give some idea.

The phylogenetic analysis is appropriate and well-thought out. Nevertheless, it would have been interesting and informative to also show the fraction of Haloferax species/strains that encode those proteins. 

Reviewer #2: Schulze et al. describe the glycoproteome of Haloferax volcanii, the most studied archaea to date. The authors made a significant contribution implementing mass spectrometry to identify peptides and proteins targeted by 2 known glycosylases from cell extracts. Among the main discoveries, authors elucidated the required presence of AglB for global N-glycosylation activity in the cell, and also the presence of Agl15-dependent N-glycosylation of proteins across different growth phases. The latter evidence suggests that Agl15 is not only important for salt-stress response but also steady-state maintenance of post-translational modifications. This method represents a new and powerful tool for future cell biology studies to map important pathways responsible for morphogenesis in haloarchaea.

Overall, the manuscript is well written and conclusions are clear and justifiable by the presented data. However, there are important controls and clarifications that are missing in the paper, especially regarding Figure 2.

Specific comments, as they show in the manuscript:

Line 86: Correct "archaealla"

Lines 117-118: I recommend avoid overinterpretation from the observation that the agl15 mutant grows at higher ODs, provided readers could misunderstand that this reflects a significantly higher cell density. Because the difference is under 10%, other factors could be contributing to absorbance at higher ODs (e.g. scattering caused by secretion of extracellular matrix). Since this is not really relevant to the core claims of the paper, it would be safer to cut this segment from the manuscript or explain the implications in the discussion section (lines 522-523).

Lines 124-129: Differences in cell morphology look rather subtle. To really state these images are representative, authors should show any form of measurements to differentiate between rods and disks. I see that the corresponding author has successfully done this in a recent work (Abdul-Halim 2020). I also think it's remarkable that the lack of any N-glycosylation barely affected the aglB shape profile. Authors should mention the fact and put in context with past papers that show S-layer defects in different salt concentrations.

Lines 172-174: I am having trouble understanding some points of the following segment:

A) How did the authors identify which protein is the SLG? Is it the ~100kDa band? Please clarify and update the panel indicating SLG

B) Even though the authors calibrated the loading of the gel, it's not clear if coomassie is the best way to semi-quantitatively compare different fractions as the dye labels differently oxidated proteins and glyco groups, which can be on the way to detect shedding itself. In order to avoid these artifacts, authors should stain gels with a quantitative staining protocol, e.g. silver or fluorescent probes.

C) Another problem with the above claim is the also observed enriched band of SLG in the aglB mutant. Using adequate staining will allow authors to get membrane:supernatant ratios to make the point there is (or not) enrichment in a specific fraction. The same measurement should be done in the glycoprotein staining.

D) Additionally, authors should clarify if different staining in 2A and 2B were performed in the same or different gels. It's my understanding that it should be done in different gels, from both biological and technical triplicates to avoid staining artifacts.

E) Why glycostainning is labeling all proteins similarly to coomassie? Authors should provide a control with cells under PNGase, Endoglycosidase or any hydrolase that would work with your cells. Likewise, I don't understand the discrepancy between glycostanning and mass spec results (that cannot detect any glycopeptides from aglB- cells). Is that related only to O-glycosylation? Please clarify in the text.

Lines 198-201: Please clarify how equal culture volumes yield a quantitative comparison between mid- and late-exponential growth phases. Higher the OD, higher the density, so different N of cells?

Lines 236-237: I don't understand how authors can infer that the vast majority of glycopeptides were identified. Was that calculated from the predicted glycoproteins? Please clarify.

Figure 3: Panels A-B relate to the total pool of peptides/proteins, while panels C-D zoom in into SLG. I think it would help the reader if panels C-D had SLG labeled directly in the figure. Additionally, I have struggled to understand whether all AglB- and Agl15-dependent glycomodifications were detected in all SLG molecules, or there was a mixed population of different modified SLG molecules. If the latter follows, it would be incredibly exciting to be discussed here whether it points out to a spatiotemporal regulation within individual cells or different profiles during shape transitions throughout growth phases.

Figure 5: I recommend replace "reveals conservation within haloarchaea" with "reveals conservation restricted to haloarchaea". Why did the authors compare to Thaumaarchaeota and not other archaeal families? Please clarify in the text.

Line 582: I believe authors have a typo. Instead of 0.03, shouldn't be 0.3 for mid-exponential growth?

Strain List: There is little information about the strains used in the study and how they were made/confirmed.

---

## [Editor Report · Decision Letter 2]

27 Apr 2021

Dear Dr Pohlschroder,

Thank you for submitting your revised Methods and Resources paper entitled "Comprehensive glycoproteomics shines new light on the complexity and extent of glycosylation in archaea" for publication in PLOS Biology. I'm handling your manuscript temporarily while my colleague Dr Paula Jauregui is out of the office. We and the Academic Editor have now assessed your responses and revisions. 

Based on our assessment, we will probably accept this manuscript for publication, provided you satisfactorily address our data and other policy-related requests.

IMPORTANT: Please supply numerical values underlying Figs 2AC, 4AB, 5, S2AB, S3, S4, S5, S6AB, and cite the location of the data clearly in the relevant Figure legends.

We expect to receive your revised manuscript within two weeks. 

*Published Peer Review History*

*Early Version*

Sincerely,

Roli Roberts

Roland G Roberts PhD

Senior Editor

PLOS Biology

on behalf of

Editor,

pjaureguionieva@plos.org,

PLOS Biology

DATA POLICY:

Many thanks for supplying raw data and scripts in PRIDE, Zenodo and Github; however, we also ask that all individual numerical values summarized in the figures and results of your paper be made available in one of the following forms:

Regardless of the method selected, please ensure that you provide the individual numerical values that underlie the summary data displayed in the following figure panels as they are essential for readers to assess your analysis and to reproduce it: Figs 2AC, 4AB, 5, S2AB, S3, S4, S5, S6AB. NOTE: the numerical data provided should include all replicates AND the way in which the plotted mean and errors were derived (it should not present only the mean/average values).

We require the original, uncropped and minimally adjusted images supporting all blot and gel results reported in an article's figures or Supporting Information files. We will require these files before a manuscript can be accepted so please prepare and upload them now. Please carefully read our guidelines for how to prepare and upload this data: https://journals.plos.org/plosbiology/s/figures#loc-blot-and-gel-reporting-requirements 

DATA NOT SHOWN?

---

## [Editor Report · Decision Letter 3]

10 May 2021

Dear Dr. Pohlschroder,

On behalf of my colleagues and the Academic Editor, Simonetta Gribaldo, I am pleased to say that we can in principle offer to publish your Methods and Resources "Comprehensive glycoproteomics shines new light on the complexity and extent of glycosylation in archaea" in PLOS Biology, provided you address any remaining formatting and reporting issues. These will be detailed in an email that will follow this letter and that you will usually receive within 2-3 business days, during which time no action is required from you. Please note that we will not be able to formally accept your manuscript and schedule it for publication until you have made the required changes.

PRESS

Thank you again for supporting Open Access publishing. We look forward to publishing your paper in PLOS Biology. 

Sincerely, 

Paula

---

Paula Jauregui, PhD 

Associate Editor 

PLOS Biology